# Integrated digital pathology and transcriptome analysis identifies molecular mediators of T-cell exclusion in ovarian cancer

Mélanie Desbois [1,7], Akshata R. Udyavar[2,7], Lisa Ryner[1,7], Cleopatra Kozlowski[1], Yinghui Guan [1], Milena Dürrbaum[2], Shan Lu[1], Jean-Philippe Fortin[2], Hartmut Koeppen[3], James Ziai[3], Ching-Wei Chang[4], Shilpa Keerthivasan[5], Marie Plante[6], Richard Bourgon[2], Carlos Bais[1], Priti Hegde[1], Anneleen Daemen[2], Shannon Turley [5] & Yulei Wang [1✉]

Close proximity between cytotoxic T lymphocytes and tumour cells is required for effective immunotherapy. However, what controls the spatial distribution of T cells in the tumour microenvironment is not well understood. Here we couple digital pathology and transcriptome analysis on a large ovarian tumour cohort and develop a machine learning approach to molecularly classify and characterize tumour-immune phenotypes. Our study identifies two important hallmarks characterizing T cell excluded tumours: 1) loss of antigen presentation on tumour cells and 2) upregulation of TGFβ and activated stroma. Furthermore, we identify TGFβ as an important mediator of T cell exclusion. TGFβ reduces MHC-I expression in ovarian cancer cells in vitro. TGFβ also activates fibroblasts and induces extracellular matrix production as a potential physical barrier to hinder T cell infiltration. Our findings indicate that targeting TGFβ might be a promising strategy to overcome T cell exclusion and improve clinical benefits of cancer immunotherapy.

---

[1] Department of Oncology Biomarker Development, Genentech, Inc., South San Francisco, CA, USA. [2] Department of Bioinformatics & Computational Biology, Genentech, Inc., South San Francisco, CA, USA. [3] Department of Research Pathology, Genentech, Inc., South San Francisco, CA, USA. [4] Department of Biostatistics, Genentech, Inc., South San Francisco, CA, USA. [5] Department of Cancer Immunology, Genentech, Inc., South San Francisco, CA, USA. [6] Laval University Cancer Research Center, Hôtel-Dieu-de-Québec, Centre Hospitalier Universitaire (CHU) de Québec, 11 Côte du Palais, Québec, QC G1R 2J6, Canada. [7]These authors contributed equally: Mélanie Desbois, Akshata R. Udyavar, Lisa Ryner. ✉email: wang.yulei@gene.com

The clinical success of cancer immunotherapies such as immune checkpoint inhibitors has revolutionized traditional cancer treatment[1]. By targeting the immune checkpoint regulators including CTLA-4 and the PD-1/PD-L1 axis, these immunotherapies promote cytotoxic killing of cancer cells by enhancing the function of effector T cells. Despite impressive efficacy demonstrated in subsets of patients with melanoma, NSCLC, urothelial bladder cancer and renal cell cancer[2–5], significant challenges still exist in this field. Dramatic and durable responses were mainly observed in subsets of patients with a preexisting T-cell immunity in tumours[6]. As such, other steps in the tumour immunity cycle may influence the effectiveness of immunotherapies based on checkpoint blockade. These include antigen presentation and T-cell priming, the capacity of tumour infiltration by functional CD8[+] T effector cells, as well as accumulation of immunoregulatory mechanisms that evolved to protect tissue integrity from exuberant immune responses[7]. Overcoming mechanisms that impede immune activation may thus enhance the potential of cancer immunotherapy.

CD8[+] T cells are the main players in eradicating cancer cells in most of the immunotherapy settings. CD8[+] T cells recognize tumour-associated antigens through the major histocompatibility complex I (MHC-I)/T-cell receptor (TCR) complex and mediate cytotoxic killing of tumour cells. Given that effective cytotoxic killing requires direct contact between CD8[+] T cells and tumour cells, it has been increasingly recognized that different CD8[+] T-cell distributions in the tumour microenvironment (TME) may elicit different responses to immunotherapy[8,9]. Several studies have also shown that the numbers of tumour-infiltrating CD8[+] T cells are associated with good prognosis in ovarian cancer[10–12]. Three basic tumour-immune phenotypes have been described previously[13], including (1) the inflamed/infiltrated phenotype, in which CD8[+] T cells infiltrate the tumour epithelium; (2) the immune-excluded phenotype, in which infiltrating CD8[+] T cells accumulate in the tumour stroma rather than the tumour epithelium and (3) the immune desert phenotype, in which CD8[+] T cells are either absent or present in very low numbers. These histologically established tumour-immune phenotypes provided a useful albeit subjective framework to profile immune contexture in solid tumours. However, it remains challenging to systematically define the tumour-immune phenotype of most cancer patients due to the highly heterogeneous and complex nature of immune cell infiltration and distribution. More importantly, the molecular features and mechanisms that shape the spatial distribution of tumour-infiltrating CD8[+] T cells are not well understood.

In this study, integrating digital pathology with transcriptome analysis in 370 ovarian tumour tissues from the ICON7 Phase III clinical trial[14], we employ a machine-learning approach to classify and molecularly characterize tumour-immune phenotypes in ovarian cancer. Using this approach, we are able to identify molecular features associated with distinct immune phenotypes. Further, our work identifies TGFβ as an important molecular mediator of CD8[+] T-cell exclusion in ovarian cancer. Our findings suggest that targeting the TGFβ pathway might be a promising therapeutic strategy to overcome T-cell exclusion from tumours and optimize responses to cancer immunotherapy.

## Results

**Infiltration of CD8+ T cells follows a continuum.** In this study, we first set out to build a set of quantitative metrics for characterizing immune phenotypes in ovarian cancer. CD8 immunohistochemistry (IHC) with a haematoxylin counterstaining was performed on 370 archival tissues (treatment-naive specimen) from a subset of patients with ovarian cancer enrolled in the ICON7 trial[14,15]. A digital image analysis algorithm was developed to quantify CD8[+] T-cell densities in the tumour epithelium versus stroma compartment. Specifically, with independent curations by a pathologist, we designed an algorithm to distinguish cells of the ovarian tumour epithelium from those of the stroma, based on the shape and size of cell nuclei from the haematoxylin staining. With a robust distinction between the tumour epithelium areas and stroma areas, this algorithm quantified the total CD8[+] T-cell count as well as CD8[+] T-cell counts per tumour epithelium and stroma area (Fig. 1a, "Methods"). To better capture and quantify the CD8 infiltration patterns, we converted these CD8 scores into polar coordinates defining two new quantitative metrics: (1) the quantity of CD8[+] T cells and (2) the spatial distribution of CD8[+] T cells ("Methods"). Next, we used these two digitally defined quantitative metrics to profile the immune phenotype of each tumour using a two-dimensional map (Fig. 1b). Representative tumours of the infiltrated, excluded and desert immune phenotypes, manually defined by a pathologist, were highlighted to validate the two digital metrics, with desert tumours having low CD8[+] T-cell quantity (R score), and excluded versus infiltrated tumours differing in the spatial distribution of CD8[+] T cells (θ score). The distinct patterns of CD8[+] T-cell distribution in digitally denoted stroma versus tumour epithelial areas of these selected tumours from Fig. 1b are illustrated in Fig. 1c. Furthermore, our results demonstrated that both total CD8[+] T-cell quantities and their spatial distribution in the TME are more on a continuum rather than discrete entities in the vast majority of tumours (Fig. 1b). These results highlighted the necessity and advantages of using our digitally devised two-dimensional quantitative metrics to define the immune phenotype of individual ovarian tumours.

**Immune-excluded tumours are associated with poor prognosis.** We next explored if a gene expression-based molecular classifier could be developed with a machine-learning approach to characterize tumour-immune phenotypes. Figure 2a summarizes the development workflow of our analysis. In this approach, transcriptome analysis was integrated with the digital pathology analysis on the same set of samples from the ICON7 trial. Using a training set of 155 for which we have digital pathology scores, we identified 352 genes whose expression can be predicted by the quantity and/or spatial distribution of CD8[+] T cells using a random forest regression model ("Methods", Supplementary Fig. 1a–c and Supplementary Data 1). Among these genes, 103 genes were associated with total CD8[+] T-cell quantity, 56 genes varied in expression by spatial CD8[+] T-cell distribution and 193 genes were associated with both total quantity and spatial distribution (Fig. 2b and Supplementary Fig. 1c). Focusing on the 159 genes that are exclusively associated with either the quantity or spatial distribution of CD8[+] T cells, we performed consensus clustering on the 155 samples and identified 6 clusters with distinct molecular profiles ("Methods", Supplementary Fig. 1d–f). These six clusters could each be readily assigned to one of the three previously defined tumour-immune phenotypes, i.e., infiltrated, excluded and desert, given their association with low versus moderate-to-high total CD8[+] T-cell quantity, or with CD8[+] T-cell enrichment in stroma versus tumour cells. Next, we built a 157-gene classifier to distinguish these three tumour-immune phenotypes, by applying the prediction analysis of microarrays (PAM) approach[16] to the training set (Supplementary Fig. 2, "Methods"). We applied this classifier to the remaining 215 tumour samples from the ICON7 collection (Fig. 2c) as an independent testing set. From the ICON7 testing set, 196 out of the 215 samples (91%) could be confidently classified, among which 64 tumours as infiltrated (30%), 44 as

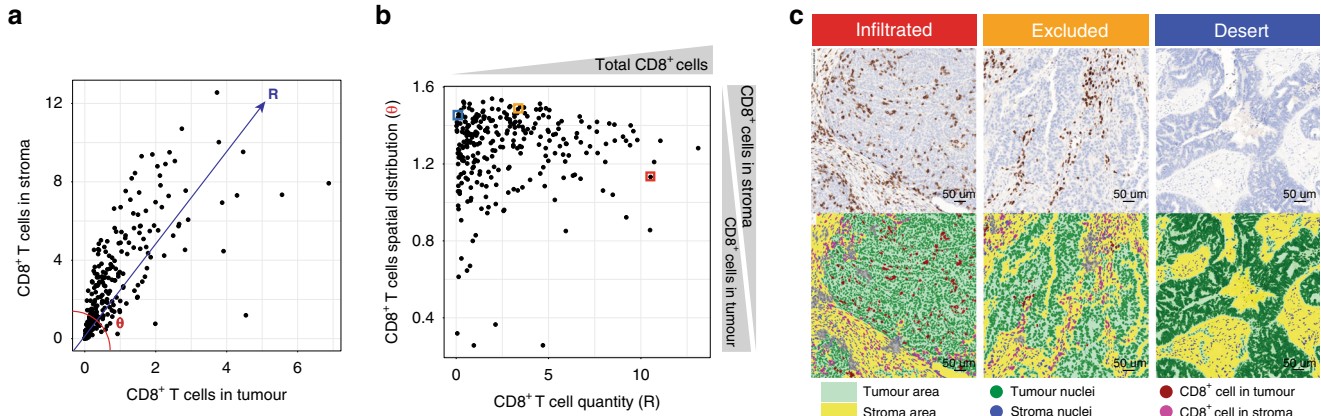

**Fig. 1 Spatial quantification of CD8$^+$ T cells with a digital image analysis algorithm. a** Digital pathology quantified infiltration of CD8$^+$ T cells in the stroma and the tumour epithelium. Scores for each compartment are displayed ($n = 155$ samples) and **b** converted into polar coordinates defining two digitally quantitative metrics to profile the immune phenotype of each tumour from the ICON7 training set ($n = 155$ samples): (1) $x$ axis, the quantities of CD8$^+$ T cells, defined as $R =$ square root [(CD8 tumour)$^2$ + (CD8 stroma)$^2$] and (2) $y$ axis, the spatial distribution of CD8$^+$ T cells, defined as $\theta =$ atan (CD8 stroma/CD8 tumour). Dots highlighted by a coloured square (blue for desert, orange for excluded and red for infiltrated) are the tumour cases displayed in **c**. **c** Example images of representative infiltrated, excluded and desert tumour-immune phenotypes, were shown to illustrate their distinct CD8$^+$ T-cell distribution in digitally denoted stroma versus tumour areas. Tumour areas (green), stroma areas (yellow), CD8$^+$ cells present in the tumour (dark-red), CD8$^+$ cells present in the stroma (pink). This digital pathology analysis has been performed on 277 tumour samples. Source data are provided as a Source Data file.

excluded (20%) and 88 as desert (41%) (Fig. 2c). CD8 IHC data and digital pathology analysis were only available for 122 out of the 215 tumour samples from the testing set. The two-dimensional metrics defining CD8$^+$ T-cell quantities and distribution for these 122 samples confirmed that the classifier assigned them to an appropriate immune phenotype (Fig. 2d, right panel). We also selected a subset of 114 samples and compared the tumour-immune phenotypes predicted by the 157-gene molecular classifier with those manually annotated by a pathologist. The results were highly concordant for excluded tumours, but the concordance dropped for infiltrated and desert tumours due to the arbitrary nature of the pathologist cut-offs defining the infiltration based on the number of cells (Supplementary Fig. 2d).

Four clinically and biologically relevant molecular subtypes, i.e., immunoreactive (IMR), mesenchymal (MES), proliferative (PRO) and differentiated (DIF), have been previously identified in ovarian cancer[17–19]. We next assessed the relationship between the tumour-immune phenotypes defined in this study and the predicted molecular subtypes based on previously developed classifier[18,19]. As shown in Fig. 2e, strong concordance was observed between the two classification schemes in both the training and testing datasets from the ICON7 study. Specifically, the IMR molecular subtype was highly enriched for the infiltrated immune phenotype, while MES tumours were highly enriched for the excluded phenotype. Desert tumours were primarily of the PRO or DIF molecular subtypes.

Finally, we found a significant association of the tumour-immune phenotypes with clinical outcome in ovarian cancer. We performed a Cox proportional hazards analysis on the dataset from 172 patients enrolled in the chemo-control arm of the ICON7 clinical trial with uniform follow-up. As shown in Fig. 2f, patients with the T-cell excluded phenotype showed significant shorter progression-free survival (PFS) as compared to patients with the infiltrated or the desert phenotype. Similarly, we demonstrated that the MES tumours, a molecular subtype that significantly overlaps with the T-cell excluded immune phenotype, also showed significantly worse PFS compared to patients with a PRO or DIF subtype. On the other hand, we did not

observe a significant difference in PFS between the infiltrated and desert immune phenotypes in our study (Fig. 2f). This may be partly due to the mixed intrinsic biology represented by the desert immune phenotype. Supporting this notion is a trending difference in PFS between the two molecular subtypes enriched in the desert immune phenotype, the DIF and the PRO subtype of ovarian cancer (Fig. 2f). Finally, we performed multivariate analysis to include in several known prognosis factors in ovarian cancer such as stage, age and debulking status. We confirmed that patients with late-stage disease (stage III and IV) and sub-optimal debulking status were significantly associated with poor prognosis in the ICON7 cohort. However, the association between excluded immune phenotype and poor prognosis remained significant even after correction of the potential effect of these known prognosis factors (Supplementary Fig. 3).

These findings highlighted the clinical relevance of the tumour-immune phenotypes and provided insights into their association with the intrinsic biological processes implicated in the molecular subtypes.

**Molecular features define distinct immune phenotypes**. We next identified key molecular features associated with the two quantitative metrics defining distinct immune phenotypes. Among the 159 genes identified in the ICON7 training set, we found that the 103 genes associated with total CD8$^+$ T-cell quantities mostly constituted a cytotoxic signature (e.g., *GZMA*, *GZMB*, *GMZH*, *CD40LG*) and served as the primary feature to distinguish the desert tumours from the infiltrated and excluded tumours (Fig. 3a). On the other hand, multiple distinct molecular features were enriched among the 56 genes associated with the CD8$^+$ T-cell spatial distribution, including antigen presentation (i.e., *TAPBP*, *PSMB10*, *HLA-DOB*), TGFβ/stromal activity (i.e., *FAP*, *TDO2*), neuroendocrine-like features (i.e., *LRRTM3*, *ASTN1*, *SLC4A4*) and metabolism (i.e., *UGT1A3*, *UGT1A5*, *UGT1A6*) (Fig. 3a). Interestingly, the infiltrated and excluded phenotypes both exhibited a cytotoxic immune cell gene signature with variable expression from medium to high but differed markedly in the expression of antigen presentation and stromal genes (Fig. 3a). Compared to the infiltrated tumours, the excluded

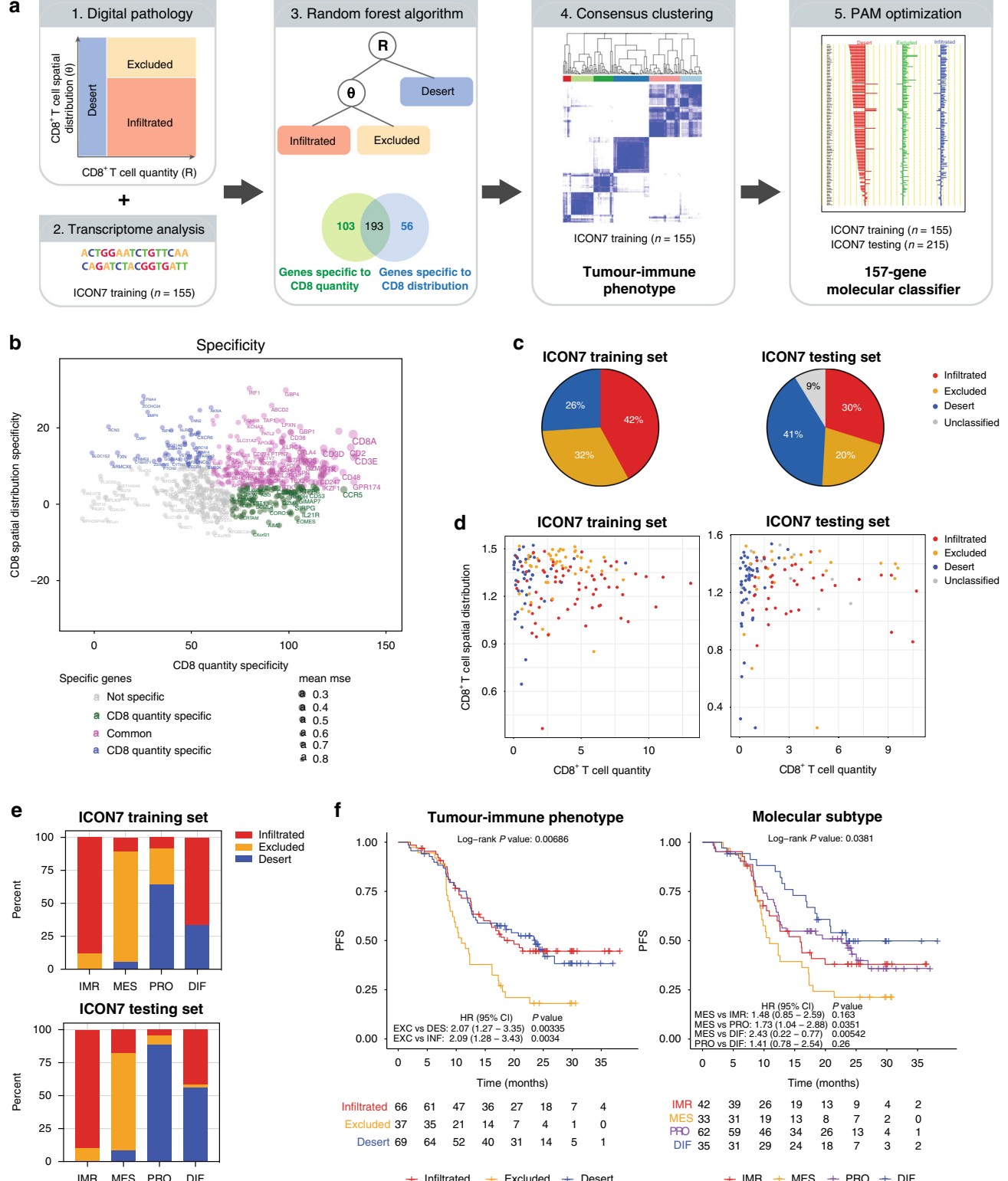

tumours featured significantly higher expression of the TGFβ-associated activated stromal genes and downregulation of antigen presentation genes. Desert tumours, on the other hand, showed a low cytotoxic gene signature as expected, but uniquely expressed metabolic genes and genes suggestive of a neuroendocrine-like state (Fig. 3a).

In order to gain a more comprehensive understanding of the biology underlying these tumour-immune phenotypes, we next performed differential pathway enrichment analysis on the full transcriptome of the 351 ICON7 samples that were classified into the distinct immune phenotypes (19 were unclassified). Based on two databases, KEGG and Hallmark, molecular pathways significantly enriched in each tumour-immune phenotype are summarized in Fig. 3b and Supplementary Fig. 4. This analysis confirmed the key biological features associated with the T-cell excluded phenotype previously identified in Fig. 3a, including the

**Fig. 2 Molecular classifier for predicting the immune phenotypes in ovarian cancer. a** A schematic illustration of the analytical workflow. **b** Genes predicting the CD8+ T-cell quantity and/or the spatial distribution metrics as identified by random forest algorithm are filtered for their specificity to each metrics (i.e., percent increase in MSE for R or θ > 3rd quantile). **c** Distribution of the predicted immune phenotypes based on gene expression in the training (n = 155) and testing (n = 215) sets of ovarian tumour samples from the ICON7 study. **d** The predicted immune phenotypes were consistently characterized by the two quantitative metrics based on digital pathology on CD8 IHC. Only samples with parallel RNA-seq data and digital pathology score of CD8 IHC are presented; (left) training (n = 155 samples) and (right) testing sets (n = 122 samples). **e** Association of the tumour-immune phenotypes and molecular subtypes in the training (n = 155 samples) and testing sets (n = 196 samples). Each bar displays the percentage of tumours of a particular molecular subtype classified as infiltrated, excluded or desert. Unclassified tumours (n = 19) were excluded from the analysis. IMR immunoreactive, MES mesenchymal, PRO proliferative, DIF differentiated. **f** Kaplan–Meier curve for progression-free survival (PFS) depicting the prognostic value of the tumour-immune phenotypes (left) and the molecular subtypes (right) in the chemotherapy arm of ICON7 (n = 172 samples). P values are generated from a Cox proportional hazard model, no multiple testing. **c–e** Source data are provided as a Source Data file.

downregulation of genes associated with antigen processing and presentation (Fig. 3c), and a strong signal for TGFβ activity with an increased expression of TGFβ ligands, a TGFβ response signature in fibroblasts (F-TBRS[8]) and an overall increase in genes indicative of active TGFβ signalling (Fig. 3d). Furthermore, pathway analysis revealed additional molecular features characterizing the distinct tumour-immune phenotypes. For example, the infiltrated tumours showed enriched interferon-alpha and gamma response (Fig. 3b, c), plausibly explaining the higher expression of antigen presentation genes in this phenotype. We also observed enrichment for the angiogenesis pathway in the immune-excluded tumours (Fig. 3b and d). For the immune desert tumours, we found that this phenotype was not only featured by the lowest expression in interferon-gamma response and antigen presentation compared to the other two tumour-immune phenotypes, it also showed a significantly downregulation of genes involved in chemotaxis (chemokine signalling) (Fig. 3c), suggesting a defect in T-cell recruitment ability. Interestingly, we also observed a slight but significant enrichment for the WNT-β-catenin signalling pathway in the desert tumours compared to infiltrated tumours. A correlation between the activation of this pathway and low expression of the T-cell gene signature has been previously reported in melanoma[20].

To evaluate in more detail which specific immune and stromal cell types are associated with a given immune phenotype, we performed a cell-type enrichment analysis using xCell[21], a gene signature-based deconvolution method, on the bulk RNA-seq datasets of ICON7 samples. The results confirmed many findings from the machine-learning and pathway enrichment analyses, including a high overall immune score in infiltrated and excluded tumours, and the highest overall stromal score in the excluded tumours (Fig. 3e). In addition, the deconvolution analysis was suggestive of significant enrichment of many immune cell types, including CD8+ T cells, regulatory T cells (Treg) and macrophages were significantly enriched in both of the infiltrated and excluded tumours compared to the desert tumours. On the other hand, the excluded tumours were specifically enriched for fibroblasts (Fig. 3e).

Lastly, genetic components, such as tumour mutation burden (TMB), neoantigen burden and high genomic instability, including microsatellite instability-high (MSI-H) and deficient mismatch repair (dMMR), have been shown to associate with increased T-cell infiltration and better responses to checkpoint inhibitors in some cancer types[22–25]. Due to limited tissue availability in ICON7 collection for genome-wide mutation profiling and HRD assessment, we performed targeted sequencing on 88 oncogenes (including both *BRCA1* and *BRCA2*) in 216 samples using a MMP-seq panel[26]. No significant association was observed between BRCA mutation status and the tumour-immune phenotypes (Supplementary Fig. 5a). To further investigate the impact of genetic components in ovarian cancer in the context of tumour-immune phenotypes, we took advantage

of the published ovarian cancer TCGA dataset for which both bulk RNA-seq and whole-exome sequencing data are available. Based on the RNA-seq data, we first predicted the tumour-immune phenotypes for 412 high-grade serous (HGS) ovarian tumour samples in the TCGA dataset by applying our 157-gene molecular classifier (Supplementary Fig. 5b and Supplementary Data 2). Our analysis revealed an overall absence of significant association between tumour-immune phenotypes and TMB, neoantigen load, dMMR or homologous recombination deficiency (HRD) in HGS ovarian cancer, with an exception that a slightly lower neoantigen load was observed in the desert compared to the infiltrated tumours (Fig. 3f). Together, our results suggest that these genetic alterations may not be a major driver of the infiltration or exclusion of CD8+ T cells in HGS ovarian cancer.

**Excluded tumours have low MHC-I and high stromal FAP expression.** We next focused our efforts on gaining more mechanistic insights into the biology underlying immune exclusion. As we have shown earlier, our integrated digital pathology and transcriptional analysis uncovered several key biological pathways and immune features underlying the T-cell excluded phenotype, including the upregulation of *FAP*, a marker of activated stroma and downregulation of antigen presentation genes. To validate these findings and distinguish which cell compartment underwent these molecular changes, we performed in situ analysis on an independent ovarian tumour collection consisting of 84 ovarian tumour samples procured from a vendor. RNA-seq transcriptome analysis was performed on these samples, and their tumour-immune phenotypes were predicted based on the 157-gene classifier developed in this study (Supplementary Fig. 5c, d). Interestingly, we observed a much higher prevalence of the excluded phenotype in the recurrent tumours compared to the primary tumours (Supplementary Fig. 5c), suggesting immune phenotypes may evolve post-chemotherapy or along with disease progression. Further validation in a larger dataset of paired primary versus recurrent tumours as well as treatment-naive versus post-chemotherapy samples is needed to confirm these findings. CD8 IHC, MHC-I IHC and FAP in situ hybridization (ISH) analyses were performed on whole slides of these tumour samples. The digital pathology algorithm developed in this study was applied to the CD8 IHC images to quantify the amount and spatial distribution of CD8+ T cells. Representative staining images of these markers from each of the three tumour-immune phenotypes are shown in Fig. 4a. A summary of all IHC or ISH scores for all samples is shown in Fig. 4b. Consistent with the findings from the ICON7 dataset, we showed that the infiltrated and excluded tumour-immune phenotypes have similar abundant quantities of CD8+ T cells by in situ analysis (Supplementary Fig. 5e), and similar CD8 mRNA expression levels by RNAseq (Fig. 4c, top). However, they differed in their distribution patterns in the tumour epithelium versus stroma area with a lower

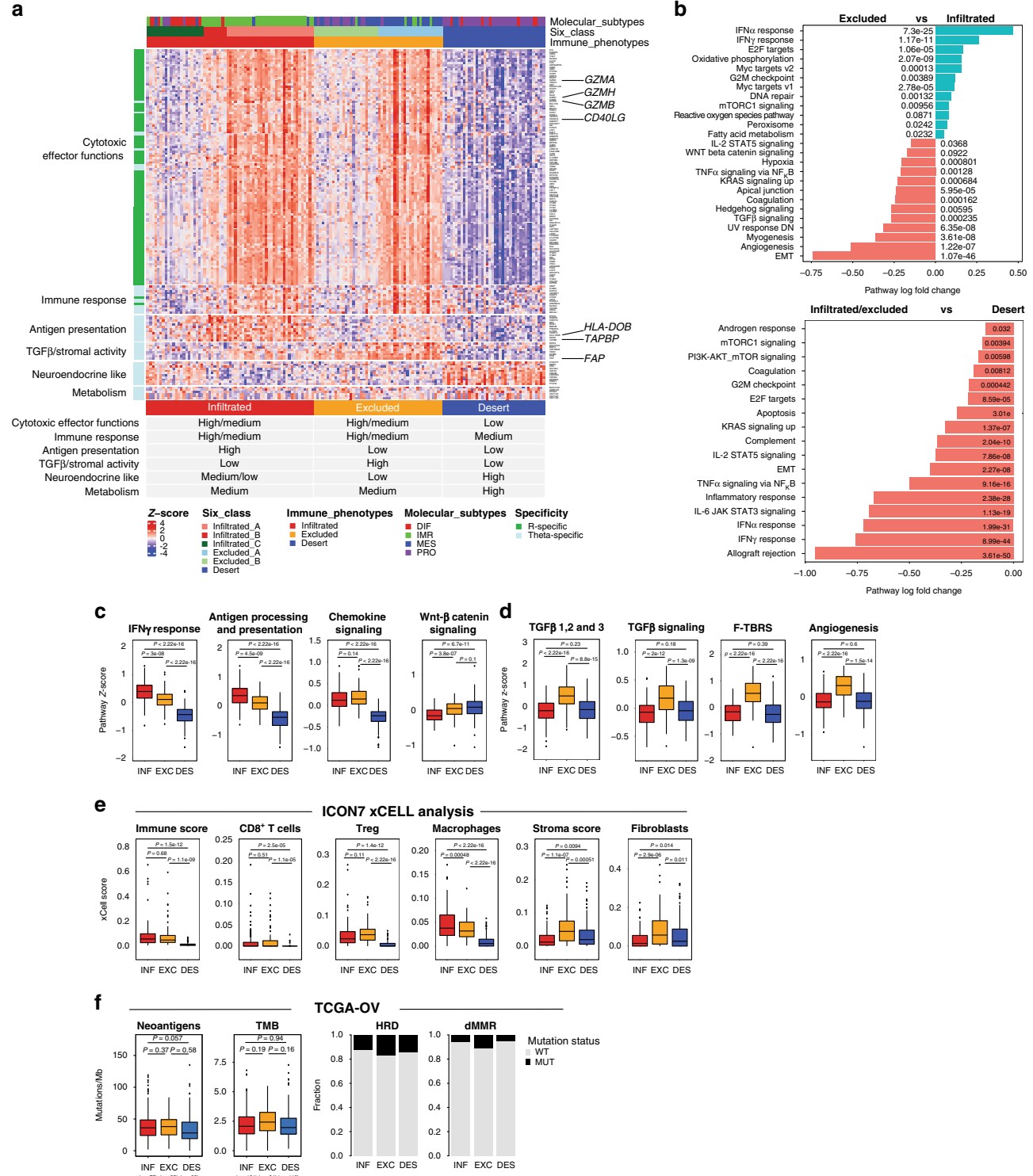

frequency of CD8$^+$ cells found in the tumour epithelium of excluded tumours (Fig. 4a, b, top).

Because neoantigen presentation and cytolytic T-cell response is governed by MHC class I molecules, which present endogenous antigens on the cell surface of almost all nucleated cells including tumour cells, we hypothesized that the infiltration and distribution of CD8$^+$ T cells might be more directly impacted by the neoantigen presenting capability via the MHC class I molecules to the CD8$^+$ T cells. Indeed, in situ analysis confirmed that the downregulation of MHC-I was associated with the excluded and a subset of desert tumours (Fig. 4b, middle row). Additionally, FAP

ISH analyses showed strong enrichment in the stroma of excluded tumours (Fig. 4b, bottom row). These findings were consistent with the results from the RNA-seq transcriptome analysis (Fig. 4c). More importantly, these in situ analyses identified specific cell compartments contributing to these observed modulations. For example, the downregulation of MHC-I in the excluded tumours was restricted to the tumour compartment. In contrast, the infiltrated tumours exhibited strong and homogenous MHC-I staining on tumour cells. On the other hand, the desert tumours exhibited both intra-tumour and inter-tumour heterogeneity in MHC-I expression.

**Fig. 3 Key molecular features characterizing distinct tumour-immune phenotypes. a** The heatmap represents the z-scored expression data of the 159 genes that associate with CD8$^+$ T-cell quantity or CD8 spatial distribution in the ICON7 training dataset (n = 155 samples). Samples are annotated on top by molecular subtypes, the six-class consensus clustering and the three-class tumour-immune phenotype. Eight genes clusters were identified. Three clusters exhibit similar biology representing cytotoxic effector functions and hence were manually pooled. The detailed gene list can be found in Supplementary Data 1. A table summarizing the biological features of the three tumour-immune phenotypes is displayed below the heatmap. **b** Enrichment analysis results for the Hallmark pathways in the entire ICON7 dataset (n = 351 samples) are depicted for (top) infiltrated versus excluded tumours, and (bottom) desert versus excluded/infiltrated tumours. Camera is the statistical method applied. **c, d** Specific pathways from the Hallmark database (IFNγ response, WNT-β-catenin signalling, TGFβ signalling and angiogenesis), the KEGG database (antigen processing and presentation and chemokine signalling) and customized gene signatures (TGFβ ligands and F-TBRS[8]) are displayed to exhibit the comparison of the three immune phenotypes (n = 351 samples, 129 infiltrated, 94 excluded and 128 desert). The pathways characterizing the infiltrated and desert phenotypes are highlighted in **c**, and those characterizing the excluded tumours in **d**. **e** Immune and stromal cell-type analysis run with xCell on bulk RNASeq from the ICON7 collection (n = 351 samples). **f** The tumour-immune phenotypes were predicted for the TCGA-OV collection (n = 378 samples) and genetic components, including neoantigen, tumour mutational burden (TMB), homologous recombination defect (HRD) and mismatch repair deficiency (dMMR) are presented for each phenotype. **c–f** Whiskers ranging from minima to maxima, median and 25–75% interquartile range (IQR) shown by boxplots. The dots are the outlier samples. Significant differences between groups were evaluated using a two-sided t test corrected for multiplicity, and the exact P values are displayed on the graphs. Source data are provided as a Source Data file.

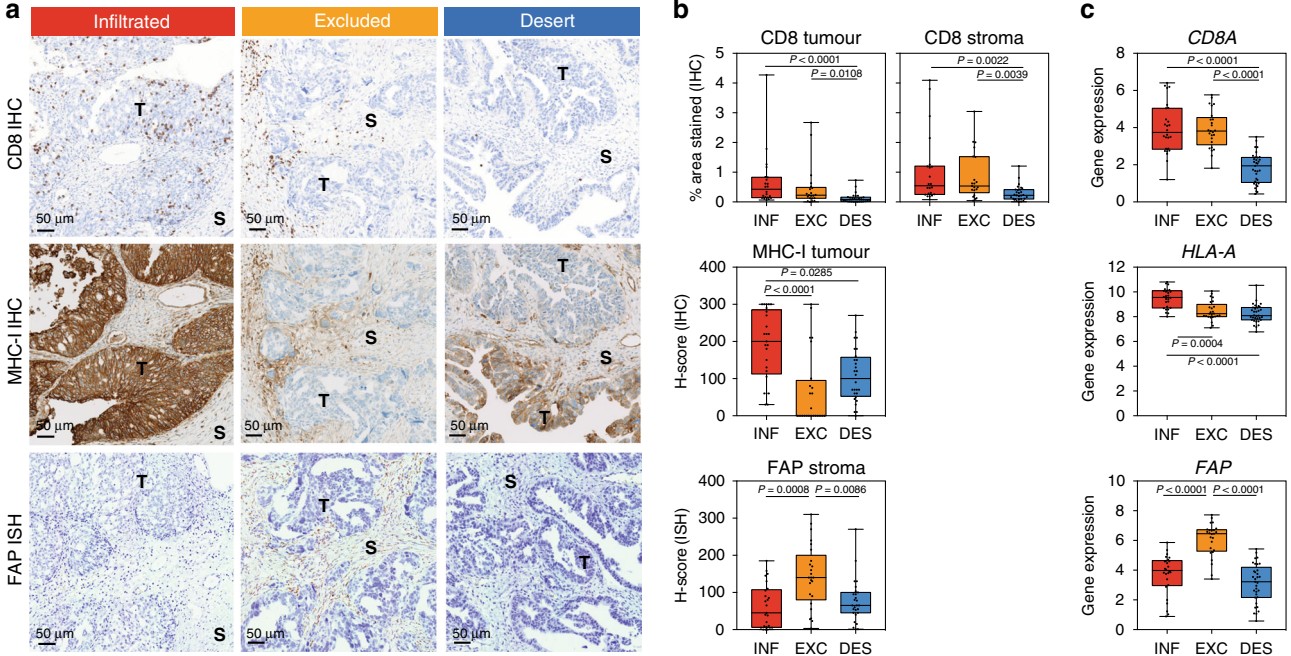

**Fig. 4 In situ validation of key features of T-cell excluded tumours.** The tumour-immune phenotypes for a vendor-procured collection (including both primary tumours and recurrent tumours, n = 84 samples) were predicted based on gene expression. The pattern of CD8$^+$ T-cell infiltration and molecular features associated with excluded tumours were validated using immunohistochemistry and in situ hybridization (ISH) on FFPE tumour tissues. **a** Representative images of CD8 IHC (top), MHC-I-IHC (middle) and FAP ISH (bottom) are shown for the three tumour-immune phenotypes. **b** Percentage of CD8 staining over tumour/stroma area (n = 72 samples), H scores for MHC-I (n = 77 samples) and FAP expression in the tumour or the stroma (n = 77 samples) were presented by the three-class tumour-immune phenotypes. **c** RNA-seq gene expression level, represented as Log$_2$(RPKM+1) for CD8A, HLA-A and FAP, is presented across the three-class tumour-immune phenotypes. **b, c** Whiskers ranging from minima to maxima, median and 25–75% IQR shown by boxplots; each dot is a tumour sample (primary tumours and recurrent tumours are pooled). The statistical significance is displayed with the exact P values on the graphs and calculated with a Kruskal–Wallis test corrected for multiple comparisons (Dunn's test). Source data are provided as a Source Data file.

This heterogeneity was reflected by intermediate H scores for MHC-I in the tumour epithelium (Fig. 4b). Together, these findings provided additional insights into potential mechanisms mediating immune exclusion, which may involve extensive crosstalk between the tumour, stroma and immune compartments.

**DNA methylation contributes to MHC-I loss in cancer cells.** We then asked what could be the mechanism of downregulation of MHC-I expression in the ovarian tumour cells. Defects of

antigen presentation machinery in tumour cells by down-regulation of MHC-I expression via genetic mutations or epigenetic suppression have been shown to represent an important mechanism of immune escape in multiple cancers[27–31]. The detection of somatic mutations in the HLA genes has been previously studied in different TCGA cohorts, including the ovarian cohort. Unlike colon and head and neck cancer, mutations in HLA genes are rare in ovarian cancer samples[32], indicating loss of MHC-I is not likely due to genetic mutations. Hence, we

investigated whether this loss is due to epigenetic regulation. To specifically detect the methylation on tumour cells, we generated DNA methylation profiles for a panel of 48 ovarian cancer cell lines using the Infinium Human Methylation 450 K Chip. A strong anti-correlation was observed between the methylation level of the promoter region of the *HLA-A* gene and its expression level (Fig. 5a), suggesting that downregulation of *HLA-A* expression in ovarian cancer is likely mediated via an epigenetic mechanism. Indeed, this hypothesis is further supported by multiple additional lines of evidence. First, we demonstrated that the observed MHC-I downregulation in ovarian cancer cells is reversible. Ovarian cancer cell lines with hypermethylation/ MHC-I[low] (OAW42) or hypomethylation/MHC-I[high] (SK-OV-3 and OVCA-420) treated with IFNγ, a cytokine well established for inducing MHC-I expression[33,34], showed increased MHC-I protein expression on the tumour cell surface (Fig. 5b and Supplementary Fig. 6a, b), supporting a reversible epigenetic mechanism rather than a hard-wired irreversible genetic modulation for MHC-I expression. More specifically, we demonstrated that in ovarian cancer cell lines with hypermethylation of *HLA-A* promoter, treatment with demethylating agent 5-aza-2'-deoxycytidine (5-Aza), a DNA methyltransferase (DNMT) inhibitor, can induce the expression of MHC-I protein at the tumour cell surface (Fig. 5c and Supplementary Fig. 6c). Collectively, these results indicated that epigenetic regulation might represent one of the important mechanisms of downregulation of antigen presentation in ovarian cancer cells to promote immune escape.

**TGFβ has multi-faceted functions in promoting T-cell exclusion.** Parallel to the downregulation of MHC-I in tumour cells, another primary feature of the T-cell excluded tumours is the upregulation of TGFβ/reactive stroma. TGFβ has been shown to downregulate MHC-I on uveal melanoma cells in vitro, and TGFβ1 null mice exhibited an aberrant expression of MHC-I and MHC-II in tissues[35–37]. Based on these reported findings, we hypothesized that TGFβ might play a direct role in downregulation of the expression of MHC-I on ovarian tumour cells. To test this hypothesis, we treated two MHC-I[high]-expressing ovarian cancer cell lines with TGFβ1. Flow-cytometry analysis revealed that TGFβ1 decreased the surface expression of MHC-I by 37.7 ± 3.2% in SK-OV-3 and 40.45 ± 14.2% in OVCA-420 compared to the untreated cells (Fig. 5d). Further, in the presence of Galunisertib, a small-molecule TGFβ inhibitor targeting the TGFβRI, MHC-I expression was restored to the untreated level (Fig. 5d).

We next evaluated if TGFβ also has a specific role in modulating fibroblasts to promote T-cell exclusion. For this, we analysed the transcriptional responses specifically induced by TGFβ treatment in primary human fibroblasts from normal ovaries, bladder and colon. We confirmed the TGFβ pathway activation by demonstrating increased phospho-SMAD2/3 in a TGFβ dose-dependent manner and pathway inhibition by galunisertib treatment (Fig. 5e). We also showed that TGFβ treatment promoted the proliferation of these primary human fibroblasts (Fig. 5f). More importantly, we identified a common 77-gene transcriptional programme specifically induced by TGFβ treatment in these human primary fibroblasts (Fig. 5g and Supplementary Fig. 6c). This transcriptional programme consists of various ECM-related genes, including collagens (*COL4A4*, *COL4A2*, *COL16A1*), ECM glycoproteins (*CTGF*, *TGFBI*, *SPARC*), proteoglycans (*BGN*, *DCN*, *VCAN*), as well as reactive stroma markers (*ACTA2*, *TNC*, *LOX*, *TIMP3*) (Fig. 5g, h). These findings suggest that TGFβ may mediate T-cell exclusion, at least in part, by creating a physical barrier via activating fibroblasts and promoting dense ECM production. In addition, these

TGFβ-activated fibroblasts might contribute to an immunosuppressive environment by producing immunomodulatory molecules, including *IL11* and *TNFAIP6* (Fig. 5g, h). Furthermore, TGFβ also increased IL-6 at the mRNA expression level and protein secretion level in the supernatant (Fig. 5i). Finally, supporting the findings from the in vitro studies, we demonstrated that many of the TGFβ induced ECM and immunomodulatory genes in vitro, were also specifically enriched in the T-cell excluded tumours in the ICON7 dataset (Fig. 5j).

Collectively, our data illuminated a multi-faceted role of TGFβ in mediating consequential crosstalk between tumour cells and cancer-associated fibroblasts to shape the tumour-immune contexture in the TME as summarized by the model presented in Fig. 5k.

## Discussion

Although histology-based tumour classification has been widely applied in the clinical setting, given the continuous nature of the distribution of tumour-infiltrating T cells, robustly classifying tumour-immune phenotypes based on only histological metrics has been challenging. In this study, we developed a digital image analysis algorithm to quantify the quantity and spatial distribution of CD8+ T cells in the TME. Coupling this digital pathology algorithm with transcriptome analysis in a large cohort of archival treatment-naive tumour tissues from the ICON7 Phase III clinical trial, we built a random forest machine-learning algorithm to classify tumour-immune phenotypes in ovarian cancer. This approach yielded a set of high-dimensional quantitative metrics to define tumour-immune phenotypes. Our study also provided the first proof of concept of classifying tumour-immune phenotypes based on a gene expression classifier. With additional optimization, validation and prospective testing, the molecular classifier developed in this study may enable systematic characterization of tumour-immune phenotypes in large clinical trials of ovarian cancer and potentially other solid tumour types as well.

Although a computational framework, tumour-immune dysfunction and exclusion (TIDE), has been reported recently for identifying factors that predict cancer immunotherapy response[38], our study represents the first study to integrate digital pathology and machine learning and provide a systematic characterization of molecular features defining distinct tumour-immune phenotypes in human cancer. As we introduced earlier, several studies have reported that a high number of tumour-infiltrating CD8+ T cells were associated with good prognosis in ovarian cancer[10–12]. Most of these studies focused on the infiltration of T cells in the tumour epithelium (the equivalent to our infiltrated immune phenotype) but possibly lumped together the rest of tumours with T cells in the surrounding stroma (the equivalent of our excluded immune phenotype) and tumours with little T cells (equivalent of desert tumours). Indeed, as shown in Supplementary Fig. 7, we observed similar results as previously reported when we combined the desert and excluded tumours (low CD8+ T cell in the tumour epithelium) in opposition to infiltrated tumours (high CD8+ T-cell infiltration in the tumour epithelium). We believe our study added an important additional layer of information, i.e., the spatial distribution of CD8+ T cells compared to the previous studies and identified that the excluded phenotype was associated with the worst prognosis in ovarian cancer. In addition, one of the key conclusions we drew from this study was that tumour-immune phenotypes should be studied and interpreted in the context of disease biology. For example, we observed that the immune desert tumours in ovarian cancer are heterogeneous and consist of two distinct molecular subtypes, the differentiated (DIF) and the proliferative (PRO) subtype. These

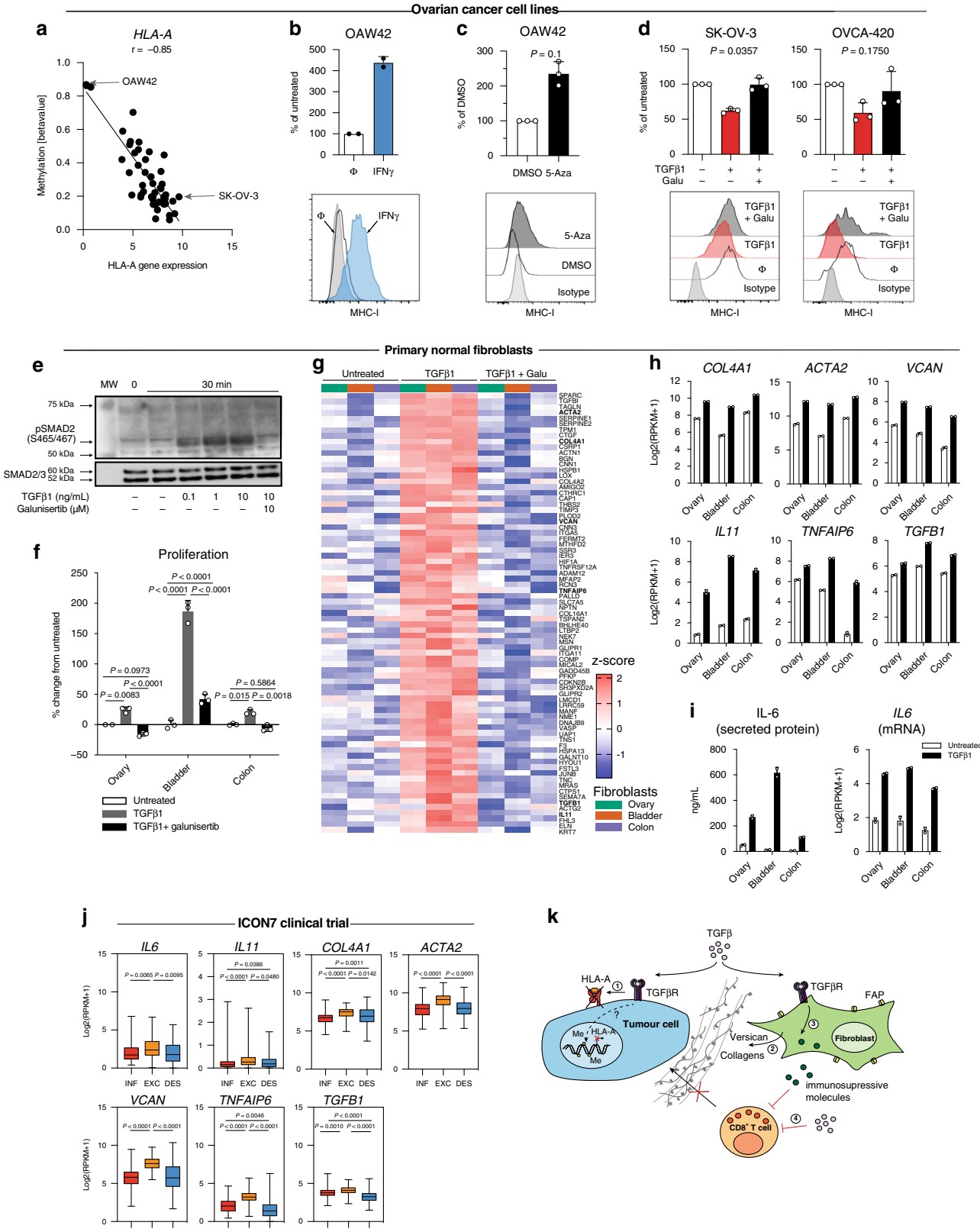

two molecular subtypes showed different clinical outcomes in previous ovarian cancer study[19] and in the ICON7 cohort of this study (Fig. 2f). Our findings call for more attention to understanding the underlying disease biology to avoid potential over-simplified classification of TIL+ and TIL- tumours with distinct biologies lumped together.

Understanding the molecular features and drivers of the T-cell spatial distribution in the TME is of major importance. There is a strong unmet need to further broaden and deepen the clinical efficacy of the immune checkpoint inhibitors. Recently, Thorsson et al. implemented a pan-cancer classification identifying six immune subtypes. Similarly, three subtypes are enriched in the

**Fig. 5 Multi-faceted role of TGFβ on ovarian cancer cells and fibroblasts. a** Correlation between *HLA-A* gene expression and its promoter methylation (beta value) for 48 ovarian cancer cell lines. **b** Change in the surface expression of MHC-I after IFNγ-treatment or **c** DNA methylation inhibitor 5-Aza treatment analysed on the OAW42 (MHC-I^low) ovarian cancer cell line by flow cytometry. **d** Analysis of surface expression of MHC-I after TGFβ1 or TGFβ1 + galunisertib (Galu) treatment on SK-OV-3 and OVCA-420 (MHC-I^high) ovarian cancer cell lines by flow cytometry. **e** Western blot on phosphorylation level of SMAD2/3 in primary bladder fibroblasts treated with different doses of TGFβ1 and Galunisertib. **f–i** Three normal human primary fibroblasts from the ovary, colon and bladder were untreated, treated with TGFβ1 or TGFβ1 + galunisertib. **f** Proliferation of the primary fibroblasts treated for 72 h. **g** Heatmap summarizing the top 77 genes specifically induced by TGFβ1. **h** Examples of genes upregulated by TGFβ1 in the primary fibroblasts. **i** Cytokines secreted by the normal fibroblasts in the presence of TGFβ1 were profiled including IL-6 protein ($n = 2$ technical replicates from one experiment) and *IL6* mRNA. **j** Expression of highlighted genes for each tumour-immune phenotype of the ICON7 collection ($n = 351$ samples). **k** Model of the role of TGFβ in promoting the exclusion of T cells in ovarian cancer. (φ) untreated and (MW) for molecular weight. The percentages of change compared to control (untreated or DMSO) are depicted in **b–d** and **f**. Data are represented as mean ± SD in **b–d**, **f**, **h**, and **i** and as a box and Whiskers plot in **j**. Whiskers ranging from minima to maxima, median and 25–75% IQR shown by boxplots. *P* values by two-tailed Mann–Whitney test in **b** and **c**, Kruskal–Wallis followed by Dunn's multiple comparisons test in **d**, two-way ANOVA in **f** and one-way ANOVA in **j** followed by Tukey's multiple comparisons tests. The results are from $n = 2$ (**b**) and $n = 3$ (**c**, **d**) independent experiments. Data presented in **e** are from one experiment with reproducible observations done after 3-h and 24-h treatment. Data depicted are from $n = 2$ (**g–i**) and $n = 3$ (**f**) biological replicates from one experiment. Source data are provided as a Source Data file.

---

ovarian cohort Wound healing (C1), IFNγ dominant (C2) and lymphocyte depleted (C4) exhibiting analogous features with our excluded, infiltrated and desert phenotypes, respectively[23]. However, our study uncovered two hallmark features characterizing the T-cell excluded tumours, including (1) loss of antigen presentation on tumour cells and (2) upregulation of TGFβ and stromal activation. More importantly, extending our previous report on the association of TGFβ with lack of response to anti-PD-L1 therapy in bladder cancer[8], this study further dissected the cell-type-specific functional role of TGFβ in mediating T-cell exclusion in ovarian cancers.

First, our mechanistic study revealed that the downregulation of MHC-I in ovarian cancer cells may be regulated by epigenetic mechanisms. Supporting this finding, we found strong anti-correlation between the *HLA-A* gene expression and promoter methylation levels. Further, we showed that IFNγ treatment as well as DNMT inhibition may overcome such epigenetic regulation and increase *HLA-A* expression in selected ovarian cancer cells. Loss of MHC-I expression regulated by epigenetic mechanisms as a result of immune pressure associated with an absence of CD8$^+$ T-cell infiltration in relapsing tumours has been previously reported in two patients with metastatic Merkel cell carcinoma treated with antigen-specific CD8$^+$ T cells and immune checkpoint inhibitors. In vitro treatment of the primary tumour cells with 5-Aza restored the expression of the MHC-I haplotype lost[31].

More importantly, we identified TGFβ as a key mediator and played multi-faceted roles in both tumour and fibroblast cells to promote T-cell exclusion. We showed that TGFβ may play a specific role in the downregulation of tumour MHC-I expression. TGFβ1 treatment decreased the surface expression of MHC-I of hypomethylated ovarian cancer cells, while TGFβ inhibition restored its normal expression level. Interestingly, previous studies have demonstrated a role for TGFβ in promoting DNA methylation through the induction of DNA methyltransferase (DNMT) expression and activity in ovarian cancer cells[39]. The molecular mechanisms underlying the regulation of tumour antigen presentation machinery by TGFβ and DNA methylation warrant additional investigations.

We also characterized another important role of TGFβ in mediating crosstalk with cancer stromal cells to promote T-cell exclusion and immunosuppression. Using human primary fibroblasts as model systems, we showed that TGFβ treatment specifically activated fibroblasts and promoted the production of ECM, which may serve as a physical barrier hindering T-cell infiltration. Furthermore, our data also suggested that TGFβ may contribute to an overall immunosuppressive TME in the T-cell-excluded tumours.

TGFβ1 treatment specifically induced immune-modulatory molecules, such as *IL6*, *IL11*, and *TNFAIP6* in human primary fibroblasts (Fig. 5g–i). Secreted in inflammatory conditions, TNFAIP6 has been reported to inhibit neutrophil migration via binding hyaluronan molecules expressed in the TME[40]. Moreover, TNFAIP6 promotes the anti-inflammatory phenotype of macrophages (M2-like), thereby contributing to the immunosuppression[41,42]. Additionally, many previous studies have reported a direct immune suppressive role of TGFβ on T cells, NK cells and dendritic cells and promoting T regulatory cell differentiation[43,44]. Altogether, our study provided additional evidence supporting a complex and multi-faceted role of TGFβ in immune modulation in cancer.

Finally, we believe our findings may have important clinical implications in the field of cancer immunotherapy. Checkpoint blockade therapies have demonstrated impressive efficacy only in subsets of patients with a pre-existing T-cell immunity[6], with the response rate even lower in ovarian cancer[45]. Therefore, there is a strong unmet need to further improve the clinical efficacy of the immune checkpoint inhibitors, and our study highlighted TGFβ as a promising target to overcome the immune escape mechanisms involved in the T-cell excluded tumours. Supporting this notion, we and colleagues have previously shown that TGFβ is associated with lack of response to anti-PD-L1 therapy in bladder cancer, especially within the T-cell excluded tumours[8]. We have also demonstrated previously that in primary and metastatic mouse colorectal cancer models, blocking TGFβ and PD-L1 signalling pathways triggered a strong T-cell infiltration in the tumour core and enhanced tumour regression and survival[8,46] supporting additional studies in breast and colorectal mouse cancer models[46,47].

It is also worth noting some of the limitations of this study. First, although we did not restrict our immune phenotyping study on HGS histology of ovarian cancer, our current study was underpowered to draw firm conclusions on non-serous histology given their relatively lower representation in our study. For example, while we found several genetic factors, such as TMB, dMMR and HRD, did not seem to be associated with different immune phenotypes in the ovarian TCGA study (HGS histology only), we cannot rule out that there may be tumour-autonomous genomic/epigenomic drivers of immune infiltration, which will differ between histological types of ovarian cancer. Second, while most of the ovarian tumour tissues analysed in this study are primary tumours, a significant portion of patients who are diagnosed with ovarian cancer have disseminated disease. Several recent reports have shown intra-tumour heterogeneity within the same individual between the metastatic and primary tissues as well as within the same

tissue[48–50]. While no significant genomic variations seem depicted between primary tumours and metastases[51], the patterns of T-cell infiltration and the composition of the TME can vary between and within patients. Hence, further investigations of immune phenotypes in extended non-serous histology and metastatic cohorts of ovarian cancer are warranted.

In summary, this study provided an in-depth and systemic characterization of the molecular features and mechanisms underlying the tumour-immune phenotypes in human ovarian cancer. This was enabled by developing a novel approach in which digital pathology was integrated with machine learning and transcriptome analysis on large ovarian tumour cohorts including the ICON7 Phase III trial. We illuminated a multi-faceted role of TGFβ in mediating consequential crosstalk between tumour cells and cancer-associated fibroblasts to shape the tumour-immune contexture in the TME. Our study highlighted the promise of targeting the TGFβ pathway as a therapeutic strategy to overcome T-cell exclusion and immune suppression and ultimately improve the patient response to cancer immunotherapy.

## Methods

**Patient cohorts and tissues**. Three hundred seventy treatment-naive patient samples with epithelial ovarian cancer from mixed histology were collected from the Phase III ICON7 clinical trial[14], and the clinical characteristics of these patients are summarized in Supplementary Fig. 8. The ICON7 protocol was compliant with good clinical practice guidelines and the Declaration of Helsinki. Approval by ethics committees was obtained at each clinical site, nationally, or both. All patients provided written informed consent. The tumour tissues were subjected to review by a pathologist to confirm diagnosis and tumour content. The cohort was divided into two sample sets for the present study: training set ($n = 155$) and testing set ($n = 215$). An independent validation collection ($n = 84$) was procured from Cureline, Inc (Brisbane, CA, USA). These include 55 treatment-naive primary tumours, and 29 paired recurrent tumours collected at disease progression post-front-line chemotherapy (25 from peritoneal cavity, 4 from the colon, liver or appendix). The ethical committee from Saint Petersburg City Clinical Oncology Hospital approved the study under the protocol CU-2010 Oncology 12152009. All patients provided written informed consent.

**Eukaryotic cell lines**. The ovarian cancer cell lines (59M, A2780, A2780ADR, Caov-3, Caov-4, COV318, COV362, COV362.4, COV413A, COV413B, COV434, COV504, COV644, DOR-13, EFO-27, ES-2, FU-OV-1, HCC630, HCC850, HEY, Hs38.T, IGROV-1, KURAMOCHI, MCAS, NIH:OVCAR-3, OAW28, OAW42, ONCO-DG-1, OV56, OV7, OVCA-420, OvCA-429, OvCA-432, OVCAR-8, OVCAR433, OVISE, OVKATE, OVSAHO, OVTOKO, PA-1, PE01, RKN, RMUG-S, SK-OV-3, TOV-112D, TOV-21G, TYK-nu, TYK-nu.CP-r) were obtained from the Genentech Cell Bank where they were authenticated by short tandem repeat profiling prior to banking and SNP fingerprinting after expansion. The human primary normal fibroblasts CCD-18-Co (colon, CRL-1459™; ATCC, Manassas, VA), HOF (ovary, #7336; ScienCell Research Laboratories, Carlsbad, CA) and Primary human bladder fibroblast (PHBF) (bladder, PCS-420-013™; ATCC) were procured from ATCC for in vitro TGFβ1 treatment.

**Immunohistochemistry and ISH assays**. Immunohistochemistry (IHC) and ISH assays were performed on 4-ʃm FFPE tissue sections. MHC-I IHC staining was performed as a single batch on the Ventana Discovery XT platform using the primary antibodies for HLA-A proteins with partial overlap with HLA-B and HLA-C proteins (Abcam #ab52922, Clone EP1395Y, diluted at 0.05 ʃg/mL (1:14,000)), the secondary anti-rabbit HRP antibodies (ThermoFisher, #65-6120, dilution 1:10,000) and a haematoxylin counterstain. CD8 IHC was performed at Histogenex on Ventana Benchmark using C8/clone 144B anti-CD8a monoclonal antibodies (Agilent Dako, #GA623). Single-plex FAP RNAscope ISH assay was designed, implemented and scored at Advanced Cell Diagnostics (Hayward, CA). The single colour probe for FAP (NM_004460.2, nt 237–1549) was pre-designed and commercially available. RNA ISH was performed using the RNAscope® 2-plex Chromogenic Reagent Kit and RNAscope® 2.0 HD Brown Reagent Kit on 4 μm formalin-fixed, paraffin-embedded (FFPE) tissue sections according to the manufacturer's instructions. RNA quality was evaluated for each sample with a dual-coloured probe specific for the housekeeping gene cyclophilin B (PPIB) and RNA polymerase subunit IIA (PolR2A). Negative control background staining was evaluated using a probe specific to the bacterial dapB gene. Only samples with an average of >4 dots per cell with the housekeeping gene probe staining and an average of <1 dot per ten cells with the negative control staining were assayed with target probes. To verify technical and scoring accuracy, reference slides consisting of FFPE HeLa cell pellets were tested for PPIB and dapB together with tissue FFPE slides. Bright-field images were acquired using a Zeiss Axio Imager M1 microscope

using a ×40 objective. H-score analysis was performed on FAP ISH and MHC-I IHC. The H-score was calculated by adding up the percentage of cells in each scoring category multiplied by the corresponding score, resulting in scores on a scale of 0–400.

**Digital pathology**. Digital pathology analysis was performed on 277 out of 370 tumours from the ICON7 collection (155 out of 155 samples from the training set and 122 out of 215 samples from the testing set had digital pathology scores) and the whole vendor-procured collection. The CD8-DAB IHC slides with a haematoxylin counterstain were scanned at ×20 magnification on a Panoramic 250 scanner (3DHistech) in MIRAX file format with 80% jpeg compression. Definiens (Munich, Germany) Developer software (v2.7.0) was used to design an algorithm to distinguish cells of the tumour epithelium from those of the stroma, using cell nuclei shape and size based on the haematoxylin signal. Once the tumour cells were identified, the immediate region surrounding those cells was defined as the tumour compartment and the rest as the stroma compartment. Within those areas, DAB$^+$ CD8 cells were counted, and the number of CD8$^+$ cells per region classified as tumour compartment, or stromal compartment was reported as tumour CD8 density, or stroma CD8 density respectively. Definiens software being discontinued, we provide a pseudo-code in the Supplementary Methods.

**Bulk RNA sequencing**. Macrodissection was performed on 370 formalin-fixed, paraffin-embedded (FFPE) tumour tissues from ICON7 as well as 84 FFPE tissues from Cureline, Inc. to enrich tumour percentage to >70%. The total RNA was purified using High Pure FFPE RNA Micro Kit (Roche Diagnostics). RNA sequencing was performed using TruSeq RNA Access technology (Illumina®). RNA-seq reads were first aligned to ribosomal RNA sequences to remove ribosomal reads. The remaining reads were aligned to the human reference genome (NCBI Build 38) using GSNAP[52,53] version 2013-10-10. To quantify gene expression levels, the number of reads mapped to the exons of each RefSeq gene was calculated using the functionality provided by the R/Bioconductor package GenomicAlignments (v1.24.0)[54]. For each cohort separately, raw counts were filtered for lowly expressed genes, whereby low expression was defined as counts per million (CPM) <0.25 in at least 10% of samples, and CPM was calculated with the cpm function in the edgeR package. Raw counts for the expressed genes were then TMM normalized based on size factors as calculated with CalcNormFactors in the edgeR package (v3.30.3). The TMM normalized counts were subsequently voom transformed with the voom function in the limma package (v3.32.6), resulting in normalized log2 CPM data. Principal component analysis (PCA) was applied to the ICON7 cohort using the normalized log2 CPM data to assess and remove any sample outliers.

Lastly, we employed an additional normalization step based on housekeeping genes, to broaden the applicability of the immune phenotype classifier to any expression platform and to tumour types beyond ovarian cancer. Housekeeping genes were identified from the pan-cancer TCGA cohort as follows: TCGA RNA-sequencing data from 11 K tumours were TMM normalized based on size factors, and voom transformed to log2 CPM data as described above. Highly expressed genes with low variance were identified in this pan-cancer expression dataset, defined as average expression exceeding 10 and variance below the 25th percentile. This revealed five candidate housekeeping genes: ACTB, ACTG1, EEF1A1, HSP90AB1 and UBC. In our ovarian cancer datasets, EEF1A1 had low expression across most samples. Hence, the ICON7 and validation datasets were each normalized to the expression of housekeeping genes ACTB, ACTG1, HSP90AB1 and UBC, by dividing the log2 CPM data by the average expression of the four housekeeping genes.

**Random forest regression**. The scores for CD8$^+$ T-cell density in tumour and stroma were strongly correlated with each other (cor = 0.74, P value <2.2e-16). To better capture and quantify the CD8 infiltration patterns, we converted these CD8 scores into polar coordinates: CD8$^+$ T-cell quantity = [square root ((CD8 tumour)$^2$ + (CD8 stroma)$^2$)] and CD8$^+$ T-cell spatial distribution = [atan (CD8 stroma/CD8 tumour)]. To identify the genes associated with these two metrics, we built a random forest regression model for each gene (gene~quantity + distribution), with standard resampling of patients but no sampling of the variables (quantity and distribution) using the randomForest package (v4.6-12). This revealed the specificity of these two metrics in predicting gene expression, for 16,944 genes in the dataset. We did not consider the bottom 25% of genes whose expression was not associated with the variables (i.e., average MSE (mean-squared error) below the first quantile). We selected genes whose expression was predicted by the quantity metric (i.e., percent increase in MSE for >3rd quantile, referred to genes associated with CD8$+$ T-cell quantity) and/or by CD8$+$ T-cell spatial distribution (i.e., percent increase in MSE for spatial distribution >3rd quantile). This resulted in 103 genes associated with CD8$+$ T-cell quantity, 56 associated with CD8$+$ T-cell spatial distribution and 193 genes common for these two metrics. Correlation analysis of these genes highlighted very similar transcriptional profiles for the 103 and 193 genes associated with CD8 quantity. For the subsequent analyses, we focused on the genes specific for these two metrics: 103 + 56 = 159 CD8$+$ T-cell quantity and distribution-associated genes, respectively.

**Consensus clustering.** Based on the 157 CD8-associated genes (159 entrez gene ids, 2 ids do not have associated gene symbol), we performed a consensus clustering on the ICON7 training set ($n = 155$) using the ConsensusClusterPlus R package (v1.42.0) with Pearson distance metric and k-means clustering with 80% patient selection and 100% feature selection. Transcriptional heterogeneity was captured well with four clusters, yet those clusters were mostly differentiated by CD8 quantity. To additionally capture CD8 distribution, we set the optimal number of clusters to six, which differentiated tumours by both CD8 quantity and distribution. The expression profile of the six clusters revealed that some clusters only differed in their cytotoxic activity, i.e., level of CD8 quantity (Fig. 3a). We therefore manually reduced the six clusters to three immune phenotypes that optimally reflected the distribution of CD8$^+$ T cells while capturing unique biological features. We labelled the immune phenotypes infiltrated, excluded and desert, given their association with low versus high CD8 quantity, and with CD8$^+$ T-cell enrichment in stroma versus tumour epithelial cells.

**PAM classification.** We used the PAMR package in R (v1.55) to derive a classifier for the prediction of the three immune phenotypes (Supplementary Data 3, R file). This classifier was built on the 157 CD8-associated genes (159 entrez gene ids, 2 ids do not have associated gene symbol), the number of necessary classifier genes ranging from 157 to 1 was evaluated, and the optimal number of genes i.e., 157 was selected corresponding to a minimal cross-validation error rate at a threshold value of 0.23. We confidently assigned a tumour to an immune phenotype when the probability for that phenotype exceeded 0.7 and was below 0.5 for the other two immune phenotypes. A tumour was otherwise considered unclassifiable.

**Gene set enrichment analysis.** The multiGSEA function (v0.13.15) with the Camera enrichment method in the multiGSEA R package was used for gene set enrichment analysis comparing different immune phenotypes in the full ICON7 collection ($n = 351$, 19 unclassified samples were excluded from the analysis), with use of the Hallmark and KEGG gene set collections from the Molecular Signature Database[55]. Pathway Z scores were calculated for each of the genesets using scores function using the multiGSEA package. Immune subset and stromal fraction enrichment analysis for ICON7 samples were done using the online xCell cell types enrichment score tool (http://xcell.ucsf.edu/).

**Mutation analysis in TCGA dataset.** Enrichment of deleterious mutations in 15 homologous recombinant deficiency (HRD)-related genes[56] and 4 dMMR genes as previously reported[57] were evaluated in TCGA-OV samples in different tumour-immune phenotypes. In addition, tumour mutation burden (TMB) and neoantigen loads were estimated in TCGA-OV samples as previously described[58]. Enrichment analysis in each tumour-immune phenotype for above-mentioned genetic features in TCGA-OV was performed using Fisher's exact test corrected for multiplicity via Benjamini–Hochberg method in R.

**Molecular subtyping of ovarian tumours.** The 100 genes that were reported in the CLOVAR signature[18] were extracted to examine the molecular subtype of a tumour. Four major clusters were identified in the ICON7 cohort based on hierarchical clustering with Euclidean distance and Ward's linkage method. By checking the testing results and up/down pattern in the original report for each gene, we could assign the identified clusters to the molecular subtypes (immunoreactive, mesenchymal, proliferative and differentiated).

**Methylation analysis of ovarian cancer cell lines.** In total, 250 ng of genomic DNA from 48 ovarian cancer cell lines were assayed using the Illumina Human Methylation 450 BeadChip platform[59]. The raw methylation data (.idat files) were read into the R software using illuminaio[60] (v0.23.2). Quality control was performed using the methylation R package minfi[61] (v1.19.0); all samples passed quality control. The methylation levels were normalized using the noob background correction and dye bias equalization methods[62] as implemented in minfi. Both procedures have been shown to perform well and to be appropriate for cancer samples[62,63]. Beta values, defined as ratios of the methylated allele intensity over the total intensity, were calculated for probes targeting CpG sites located between −1000 bp and +1000 bp from the transcription start site of the HLA-A gene.

**In vitro experiments on ovarian cancer cell lines.** SK-OV-3 and OVCA-420 (MHC-I$^{high}$), and OAW42 (MHC-I$^{low}$) ovarian cancer lines were cultured in complete culture media (RPMI-1640 + 10% FBS). The cells were plated at 12,500–100,000 cells/well in a six-well tissue culture plate and complete culture media. After 24 h, the cells were starved overnight in DMEM high glucose medium without FBS. Next, the starving media was replaced with media only (DMEM + 2% FBS), 10 ng/mL rhTGFβ1 (Cat #PHG9204, ThermoFisher, CA), 10 ng/mL rhTGFβ1 + 10 μM Galunisertib (Cat #S2230, SelleckChem, TX) or 5 ng/mL recombinant IFNγ (Cat #554617, BD Biosciences, CA) for 96 h at 37 °C. Cells were then stained and analysed by flow cytometry. The percentage of untreated was calculated using this formula: [geometric mean fluorescence intensity (IFNγ-treated cells)/geometric mean fluorescence intensity (untreated cells)] × 100. In order to see if MHC-I expression can be regulated by methylation,

MHC-I$^{low}$ OAW42 cell line was plated at 250,000 cells/dish in a 10-cm dish and serum-starved as described above for TGFβ1 treatment. In all, 10 μM of 5-Aza-2′-deoxycytidine (5-Aza, Cat #A2385, Sigma-Aldrich) demethylating agent in culture media was used to treat OAW42 for 96 h prior to FACS analysis. Media was half-replenished with fresh 5-Aza 48 h after treatment to keep concentration consistent.

**In vitro experiments on normal fibroblasts.** The primary normal fibroblast PHBF (Bladder), CCD-18Co (Colon) and HOF (Ovary) were serum-starved overnight before treatment with media only (untreated), 10 ng/mL rhTGFβ1 or 10 ng/mL rhTGFβ1 + 10 μM Galunisertib for 24 h, and the total RNA was extracted for RNA-seq analysis as previously described[8]. To detect IL-6 protein in the supernatant, cells were treated for 48 h with recombinant human TGFβ1. After the 48-h timepoint, the supernatant was collected and analysed by Luminex using the Millipore kit. For the proliferation assay, PHBF, CCD-18Co, HOF were plated at 3000 cells/well in a 96-well culture flat bottom plate for immuno-fluorescence assays (Corning, #3917) overnight. Cells were then cultured for 72 h in DMEM high glucose + 1% FBS with the indicated concentration of TGFβ1 with or without Galunisertib. Next, CellTiter-Glo® reagents (Promega, #G7570) were added to each well, and the luminescence signal was read with a microplate reader.

**p-SMAD2/3 western blot assay.** PHBF cells were plated at 100,000 cells/well in a 24-well cell culture plate overnight, serum-starved for 24 h and then cultured in serum-free DMEM with the indicated concentration of TGFβ1 with or without Galunisertib for 30 min. Cells were lysed in protein lysis buffer containing T-PER tissue protein extraction reagent (ThermoFisher, #78510), cOmplete™ Protease Inhibitor Cocktail (Sigma-Aldrich, #11697498001) and PhosSTOP™ phosphatase inhibitor cocktails (Sigma-Aldrich, #4906845001). Total protein was diluted and normalized to 0.5 μg/μL with 4× LDS sample buffer (ThermoFisher, #84788). In total, 10 μg of total protein was loaded into each well of a NuPAGE 4–12% Bis-Tris Midi Gel (Invitrogen), followed by protein transfer from the gel to the membrane using Trans-Blot Turbo (Bio-Rad). The Phospho-Smad2 was first revealed following the general protocol western blot from Bio-Rad. Briefly, the membrane was blocked for 1 h, incubated with Phospho-Smad2 antibodies overnight at 4 °C (Ser456/467, 1:200, Cell Signaling #3108, clone 138D4), washed and incubated with secondary antibodies goat anti-rabbit HRP-linked (Cell Signaling, #7074, dilution: 1:15,000). To analyse the total Smad2/3, the membrane was stripped and then incubated with Smad2/3 antibodies (Cell Signaling # 8685, clone D7G7, dilution 1:1000).

**Flow-cytometry analysis.** Before staining, Fc receptors were blocked for 10 min at room temperature using FcR blocking reagent human (Cat #130-059-901, Miltenyi Biotec, CA). Cells were stained during the blocking step with the LIVE/DEAD™ Fixable Near-IR Dead Cell (Cat #L10119, Invitrogen, CA). Then, cells were incubated at room temperature for 15 min with anti-human HLA-ABC-PE (Cat #560168, BD Biosciences, clone DX17, dilution 1:10) or isotype control mouse IgG1|-PE (Cat #556650, BD Biosciences, dilution 1:10) antibodies, washed and samples were acquired on BD LSRFortessa™ flow cytometer.

**Reporting summary.** Further information on research design is available in the Nature Research Reporting Summary linked to this article.

## Data availability

The following datasets generated during and/or analysed during the current study from ICON7 Phase 3 trial are publicly available: images of CD8 IHC and associated digital pathology outputs (EMPIAR: https://www.ebi.ac.uk/pdbe/emdb/empiar/, accession #EMPIAR-10512; raw RNA-sequencing data and clinical data (PFS) (the European Genome-Phenome Archive, accession number EGAS00001003487). All other data are available in the article and supplementary information or from the corresponding author upon reasonable request. The TCGA database is accessible at https://gdc.cancer.gov/ and the molecular signature database at https://www.gsea-msigdb.org/gsea/msigdb. Source data are provided with this paper.

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

## Acknowledgements

We thank the ICON7 Translational Research Sub-group, the University of Leeds for their work on the coordination of samples from the ICON7 trial, Ashi Malekafzali for helping with the acquisition of clinical tumour tissue samples; Erica Schleifman and Teiko Sumiyoshi for assistance with the FFPE tissue processing and RNA and DNA preparation; Roderick Pata, Carmina Espiritu and Jian Jiang who helped with sectioning tissues, Shari Lau who performed the IHC; Raj Jesudason and Kathryn Mesh who assisted with the digital analysis; Amy Lo and Rafael Cubas for their support of the study, Ivette Estay for managing the RNA extraction from ICON7 trial and all the patients who participated in the trial. This study was supported by Genentech/Roche.

## Author contributions

M.C.D., L.R., A.D. and Y.W. wrote the paper with inputs from all other authors. Y.W. conceived, designed and supervised the study. M.C.D., L.R., Y.G. and S.L. acquired or contributed to data acquisition; analysed and interpreted data. A.U., M.D, C-W.C. and J-P.F. performed the computational or statistical analysis. A.D. and R.B. supervised the computational analysis. M.P. contributed to the collection of clinical samples. C.K. developed the digital pathology algorithm. H.K. and J.Z. analysed tissue images. S.K. provided technical support. C.B., P.H., A.D. and R.B. helped conceptualize the study, optimized the method and provided resources. All authors read and agreed on the final version of the paper.

## Competing interests

M.C.D., L.R., A.U., C.K., Y.G., M.D., S.L., J-P.F., H.H., J.Z., C-W.C., S.K., C.B., P.H., A.D., R.B., S.T. and Y.W. declare that they are Genentech/Roche employees. L.R., A.U., C.K., Y.G., S.L., J-P.F., H.H., J.Z., C-W.C., S.K., C.B., P.H., A.D., R.B., S.T. and Y.W. declare that they hold Roche stocks. The remaining authors declare no competing interests.
