## [Peer Review File · Nature Communications]

Reviewers' comments:

Reviewer #1 (Immune signatures in tumors)(Remarks to the Author):

Using ovarian cancer as a case study, the authors demonstrate the value of combining digital pathology with molecular profiling to extract key biological insights from the tumor microenvironment that would otherwise remain elusive using either approach alone. The authors observed two key factors associated with T cell exclusion in CD8 T cell rich ovarian tumors: reduced antigen presentation and increased TGF-beta production. The authors followed up these observations with experiments to show that TGF-beta signaling in tumor cells and surrounding stromal cells may in fact contribute to CD8 T cell exclusion via reduced class I antigen presentation in tumor cells and activation of fibroblasts which includes up-regulation of ECM related genes that help create a physical barrier to keep CD8 T cells out. Their results provide support for inhibiting TGF-beta signaling as a means to overcome resistance to immunotherapy in T cell excluded ovarian tumors.

Overall recommendation is accept conditioned on addressing the following concerns:

Major Comments

1. It is not clearly explained why the authors:

- a. Focus on MHC class I-based antigen presentation in experiments when their initial results (see Figure 3a) do not specifically point to MHC class I based antigen presentation but both class I (TAPBP) and class II-based antigen presentation (HLA-DOB)
- b. Why they choose HLA-A as a proxy for MHC class I antigen presentation vs B2M, which is a central component of all MHC class I molecules (HLA-A, B and C) and can also be potentially targeted (See publication PMIDs: 28287290, <https://www.nature.com/articles/s41467-017-01062-w>).

2. In the "Bulk RNA-sequencing" methods section the authors' description of how they filtered, normalized and transformed RNA-seq data does not make a great deal of sense. They describe first converting raw counts to counts per million (CPM) and filtered these for low-expressed genes. They then used the resulting CPM data to perform TMM normalization and then limma-voom transformation which is not right. TMM normalization should be performed on raw counts and voom transformation should be performed on the TMM normalized counts (the voom transformation is a $\log\text{CPM} + 0.5$ transformation of these counts). To follow a more state-of-the-art approach, the authors should not first convert counts to CPM. They should instead filter low-expressed raw counts using the edgeR filterByExpr function, followed by TMM normalization and voom transformation on the resulting counts.

3. Also, in the "Bulk RNA-sequencing" methods section the authors indicate that the RNA-seq raw counts from all the samples of the ICON7 trial and possibly also the independent validation cohort were filtered, normalized, and transformed together using edgeR TMM and limma voom prior to machine learning (ML) regression. This methodology leaks test set information into the training set in two ways: 1) during the filtering of low-expressed genes step where test sample gene expression information is used to determine which training genes to filter out or not, and 2) in the TMM normalization since this method uses information across all samples (i.e. train and test) to perform normalization. Leaking of test sample information likely biases the prediction performance estimates mentioned in the paper. The correct way to do this in an ML setting is to filter out low-expressed genes and perform TMM+voom only on the training set, save the genes that were filtered out and the relevant normalization settings, and finally use these settings to apply filtering and normalization on the test set before prediction with the trained Random Forest model. See for

example the R package MLSeq where this is done.

Minor comments/Typos/grammatical mistakes:

1. The resolution of Figure 3a should be improved. It is hard to read the genes listed on the heatmap.
2. Checkpoint blockades have demonstrated impressive efficacy in only subsets of patients with a pre-existing T cell immunity⁶, with the response rate is even lower in ovarian cancer even lower
3. Figure 5a, caption:
The graph depicts an anti-correlation between the HLA-A gene expression ($\text{Log}_2(\text{RPKM}+1)$) and his promoter methylation (beta value) for 48 ovarian cancer cell lines.

Reviewer #2 (Ovarian cancer, TGFb)(Remarks to the Author):

This is a detailed report which looks at tissue from ICON 7, and systematically evaluates transcriptomic analysis to investigate mechanisms of T cell exclusion, and attempts to explain impact of this on outcome. It further investigates the biologic basis of effect on different ovarian cancer cell lines, with detailed mechanistic correlation showing effect of TGFb orchestrating T cell exclusion through effect on micro-environment, ECM and fibroblasts, with reversal using TGFb inhibitors.

Genetic alterations thought not to be be major driver of T cell exclusion but downregulation of MHC-I linked to epigenetic changes - demethylation in tumour cells, with restoration of MHC-I using epigenetic modulators or IFN. TGFb decreases MHC-I and levels of MHC-I restored with TGFb inhibitors.

The study also links tumour immunophenotype using a gene expression classifier. These findings portray a potentially elegant explanation for immune exclusion in the microenvironment and the resultant effects on response to therapy and outcome.

There are some key considerations that need some clarification.

1] The pathologic selection from ICON7 and TCGA relate to high grade serous subtype, and authors confirmed this selection. However, some of the cell lines selected for further evaluation do not seem to be restricted to serous lines, and so would like the authors to comment if the inferences drawn still apply.

PA-1: ? Teratocarcinoma

TOV112D - ? Endometrioid

COV434 - granulosa

However, SMARCA4 mutations are a feature of small cell ovarian of hypercalcemic type - and dual SMARCA4 and A2 mutations define SCCOHT cell lines -some of the ones cited in this paper have in published reports been thought to have been misclassified. It is essential that the conclusions are based on work done in HGS lines only. The epigenetic work and inhibition by EZH2 should be re-evaluated.

2] Some detail about the ICON7 patient demographics/treatment outcome for context would be helpful. Please confirm if the specimens were from only control patients?

3] Were the ICON7 experimental arm patients with bev evaluated?

4] Is there any information on BRCA status and HRD of the ICON7 samples used and was impact of this assessed

5] Were there any m/v analysis done to look at impact of immune phenotype in relation to stage/residual disease on PFS and its potential confounding effect on Fig 2f. Was there any influence of immune phenotype on stage etc.

Reviewer #3 (Ovarian cancer, pathology)(Remarks to the Author):

1) Based on CD8+ T cell quantity and epithelial distribution ovarian cancer cases were classified into infiltrated, excluded and desert. The gold standard for this categorization is unclear. The associations of these categories with survival (Fig. 2f) make no sense: it has been consistently shown that the presence and quantity of T cell infiltration is positively associated with survival e.g. PMID: 29049607. Hence, the fact that infiltrated and deserted tumors (Fig 2f) have the same outcome undermines the credibility of the proposed classification, which is the basis of this manuscript.

2) Because of limited access to CD8 IHC slides, the rationale of developing a gene expression classifier is stated. My question would be: is it not easier to fix the logistics and get CD8 IHC slides? What is the accuracy (and sensitivity/specificity) of the gene expression surrogate (because if it is not >95% it may not be sufficient in the era of precision medicine)?

3) The quality of the pathology review of ICON7 is questionable: Fig 3F shows a relevant number of cases with dMMR, which are endometrioid carcinoma and NOT high-grade serous. This mix will confound analysis, e.g. high-grade serous molecular subtypes were assigned to endometrioid carcinomas etc.

4) The studied cell lines were not carefully chosen: e.g. OAW42 and SKOV3 are unlikely high-grade serous (PMID: 23839242).

5) Relevant ovarian cancer literature is not cited, e.g. TGFb is not novel, there are already reviews, e.g. 31091744. No citation of any CD8 study e.g. PMID: 29049607

6) line 79 CD8 IHC on N=370 from ICON7 versus line 124 CD8 IHC data for 122 out of the 215 samples?

7) The lack of consistency between "primary" putative ovarian site and "metastases" putative peritoneal site is concerning (Fig. S4B). Depending whether a tumor sample is taken from the ovary versus peritoneum, the frequency of 'excluded' will increase from 13% to 62%. The results need to be controlled for the sampling site.

Desbois, Udyavar, Ryner et al.

Responses to Reviewers' Comments

Reviewer #1 (Immune signatures in tumors):

Using ovarian cancer as a case study, the authors demonstrate the value of combining digital pathology with molecular profiling to extract key biological insights from the tumor microenvironment that would otherwise remain elusive using either approach alone. The authors observed two key factors associated with T cell exclusion in CD8 T cell rich ovarian tumors: reduced antigen presentation and increased TGF-beta production. The authors followed up these observations with experiments to show that TGF-beta signaling in tumor cells and surrounding stromal cells may in fact contribute to CD8 T cell exclusion via reduced class I antigen presentation in tumor cells and activation of fibroblasts which includes up-regulation of ECM related genes that help create a physical barrier to keep CD8 T cells out. Their results provide support for inhibiting TGF-beta signaling as a means to overcome resistance to immunotherapy in T cell excluded ovarian tumors.

Overall recommendation is accept conditioned on addressing the following concerns:

Authors' response:

We highly appreciate the reviewer's constructive feedback. Below we have addressed the reviewer's comments point by point. We have also made significant changes/additions in the revised manuscript to further improve the clarity of our work.

Major Comments

1. It is not clearly explained why the authors:
 - a. Focus on MHC class I-based antigen presentation in experiments when their initial results (see Figure 3a) do not specifically point to MHC class I based antigen presentation but both class I (TAPBP) and class II-based antigen presentation (HLA-DOB)

Authors' response:

As the reviewer correctly pointed out, our initial transcriptome analysis indeed identified a group of genes involved in both class I and class II antigen presentation, i.e. TAPBP, HLA-DOB or PSMB10 (Figure 3a), that were associated with distinct tumor-immune phenotypes. For the *in situ* analysis by immunohistochemistry, we prioritized the MHC class I analysis over MHC class II based on the following considerations:

Desbois, Udyavar, Ryner et al.

- 1) Recent work has highlighted the importance of cancer-specific neoantigens in determining cytolytic T cell activity as well as predicting efficacy of immune checkpoint inhibition. A critical step in neoantigen presentation and cytolytic T cell response is governed by MHC class I molecules (HLA-A, -B, -C), which presents endogenous antigens on the cell surface of almost all nucleated cells including tumor cells.
- 2) Downregulation of HLA genes via genetic mutation, epigenetic or transcriptional regulation have been indicated as a potential mechanism of immune evasion and have been associated with resistance to immune checkpoint inhibitors. Furthermore, HLA downregulation, characterized by immunohistochemistry or monoclonal antibodies, has been found to be prevalent across a range of cancer types and has also been linked to poor outcome (Campoli and Ferrone, 2008; Hicklin et al., 1999; Hiraki et al., 2004; Mehta et al., 2008). Loss of either the maternal or paternal HLA haplotype may also impact upon the efficacy of immunotherapy.
- 3) The expression of MHC class II molecules is typically restricted to the professional antigen-presenting cells (APCs) such as dendritic cells (DC) or macrophages. Exogenous antigens that are internalized by phagocytosis are primarily presented on MHC II to CD4+ T cells. Although CD4+ T cells are important cells involved in the anti-tumour immunity by helping CD8 and humoral responses, we hypothesized that the infiltration and distribution of CD8+ T cells may be more directly impacted by the neoantigen presenting capability via the MHC class I molecules to the CD8 T cells. However, we do not rule out that we would have similar observations if we can develop a robust MHC class II IHC assay for in situ validation studies in the future.

In the revised manuscript, we now further clarified these points by adding this statement

Results (page 13)

“Because neoantigen presentation and cytolytic T cell response is governed by MHC class I molecules, which present endogenous antigens on the cell surface of almost all nucleated cells including tumor cells, we hypothesized that the infiltration and distribution of CD8⁺ T cells may be more directly impacted by the neoantigen presenting capability via the MHC class I molecules to the CD8⁺ T cells. Indeed, in situ analysis confirmed that the downregulation of MHC-I was associated with the excluded and a subset of desert tumours (Fig. 4b, middle row).”

b. Why they choose HLA-A as a proxy for MHC class I antigen presentation vs B2M, which is a central component of all MHC class I molecules (HLA-A, B and C) and can also be

Desbois, Udyavar, Ryner et al.

potentially targeted (See publication PMIDs: 28287290, <https://www.nature.com/articles/s41467-017-01062-w>).

Authors' response:

Our choice of antibody (Abcam, clone EP1395Y, Ref#ab52922) for the MHC class I IHC assay was based on the following considerations:

- 1) We chose HLA over B2M as a proxy for MHC class I presentation because we preferred to assess the cell surface expression of MHC class I using a transmembrane protein such as HLA molecules. In addition, studies in endometrial carcinoma have shown that the loss of HLA-A, B or C is more frequent than the loss of B2M. B2M loss is accompanied by the loss of HLA heavy chain (de Jong et al., 2012).
- 2) Although EP1395Y was a HLA-A specific antibody according to the manufacturer's claims, it does have overlap with HLA-B and HLA-C. This is consistent with our own in-house benchmarking data: When we compared the staining patterns of EP1395Y with an antibody that covered HLA-B and HLA-C (HC10) in a panel of tumor tissues and cell pellets, similar staining patterns were observed in the vast majority of samples (example images below).

- 3) Finally, EP1395Y antibody has been validated as a good surrogate for detecting cell surface MHC class I expression in several published studies. For example (Ritter et al., 2017; Wallin et al., 2016).

In the revised manuscript, we now added further clarification in the online methods section (page 22):

“MHC-I IHC staining was performed as a single batch on the Ventana Discovery XT platform using the primary antibodies for HLA-A proteins with partial overlap with HLA-B and HLA-C proteins (Abcam #ab52922, Clone EP1395Y, diluted at 0.05 $\mu\text{g}/\text{mL}$)”

2. In the “Bulk RNA-sequencing” methods section the authors’ description of how they filtered, normalized and transformed RNA-seq data does not make a great deal of sense. They describe first converting raw counts to counts per million (CPM) and filtered these

Desbois, Udyavar, Ryner et al.

for low-expressed genes. They then used the resulting CPM data to perform TMM normalization and then limma-voom transformation which is not right. TMM normalization should be performed on raw counts and voom transformation should be performed on the TMM normalized counts (the voom transformation is a $\log\text{CPM} + 0.5$ transformation of these counts). To follow a more state-of-the-art approach, the authors should not first convert counts to CPM. They should instead filter low-expressed raw counts using the edgeR filterByExpr function, followed by TMM normalization and voom transformation on the resulting counts.

Authors' response:

We would like to thank the reviewer for the very thorough review of our manuscript and identified this inaccurate description of our data processing method. After careful review of our codes for "bulk RNA-sequencing", we confirmed that the actual process we used in this study is following: 1) we first filtered raw counts for lowly expressed genes, 2) followed by TMM normalization and then voom transformation. To note, we employed an additional normalization step based on housekeeping genes that was not described in the initial submission.

In the revised manuscript, we now have updated the online methods paragraph as follow (pages 24-25):

"Bulk RNA-sequencing

Macrodissection was performed on 370 formalin-fixed, paraffin-embedded (FFPE) tumour tissues from ICON7 as well as 84 FFPE tissues from Cureline, Inc. to enrich tumour percentage to greater than 70%. Total RNA was purified using High Pure FFPE RNA Micro Kit (Roche Diagnostics). RNA sequencing was performed using TruSeq RNA Access technology (Illumina®). RNA-seq reads were first aligned to ribosomal RNA sequences to remove ribosomal reads. The remaining reads were aligned to the human reference genome (NCBI Build 38) using GSNAP^{43,44} version 2013-10-10. To quantify gene expression levels, the number of reads mapped to the exons of each RefSeq gene was calculated using the functionality provided by the R/Bioconductor package GenomicAlignments⁴⁵. For each cohort separately, raw counts were filtered for lowly expressed genes, whereby low expression was defined as counts per million (CPM) <0.25 in at least 10% of samples, and CPM was calculated with the cpm function in the edgeR package. Raw counts for the expressed genes were then TMM normalized based on size factors as calculated with CalcNormFactors in the edgeR package. The TMM normalized counts were subsequently voom transformed with the voom function in the limma package, resulting in normalized log₂ CPM data. Lastly, we employed an additional normalization step based on housekeeping genes, to broaden the applicability of the immune phenotype classifier to any

Desbois, Udyavar, Ryner et al.

expression platform and to tumor types beyond ovarian cancer. Housekeeping genes were identified from the pan-cancer TCGA cohort as follows: TCGA RNA-sequencing data from 11K tumors were TMM normalized based on size factors, and voom transformed to log2 CPM data as described above. Highly expressed genes with low variance were identified in this pan-cancer expression dataset, defined as average expression exceeding 10 and variance below the 25th percentile. This revealed five candidate housekeeping genes: ACTB, ACTG1, EEF1A1, HSP90AB1, and UBC. In our ovarian cancer datasets, EEF1A1 had low expression across most samples. Hence, the ICON7 and validation datasets were each normalized to expression of housekeeping genes ACTB, ACTG1, HSP90AB1, and UBC, by dividing the log2 CPM data by the average expression of the four housekeeping genes. Principal component analysis (PCA) was applied to the ICON7 cohort using the normalized log2 CPM data to assess and remove any sample outliers.

3. Also, in the “Bulk RNA-sequencing” methods section the authors indicate that the RNA-seq raw counts from all the samples of the ICON7 trial and possibly also the independent validation cohort were filtered, normalized, and transformed together using edgeR TMM and limma voom prior to machine learning (ML) regression. This methodology leaks test set information into the training set in two ways: 1) during the filtering of low-expressed genes step where test sample gene expression information is used to determine which training genes to filter out or not, and 2) in the TMM normalization since this method uses information across all samples (i.e. train and test) to perform normalization. Leaking of test sample information likely biases the prediction performance estimates mentioned in the paper. The correct way to do this in an ML setting is to filter out low-expressed genes and perform TMM+voom only on the training set, save the genes that were filtered out and the relevant normalization settings, and finally use these settings to apply filtering and normalization on the test set before prediction with the trained Random Forest model. See for example the R package MLSeq where this is done.

Authors’ response:

We understand our statement in the “Bulk RNA-sequencing” methods section led to a misinterpretation and would like to take the opportunity here to clarify our method. As point out by the reviewer, we normalized each cohort (ICON7, TCGA and vendor-procured cohorts) separately. To note, the training and testing sets for ICON7 were normalized together. However, the impact here is not expected to be substantial.

Desbois, Udyavar, Ryner et al.

In the revised manuscript, we now have updated the Methods section as following (page 24-25): “For each cohort separately, raw counts were filtered for lowly expressed genes, whereby low expression was defined as counts per million (CPM) <0.25 in at least 10% of samples, and CPM was calculated with the cpm function in the edgeR package.”

Minor comments/Typos/grammatical mistakes:

1. The resolution of Figure 3a should be improved. It is hard to read the genes listed on the heatmap.

Authors’ response:

In the revised manuscript, we improved the resolution of the Figure 3 and updated the Supplement table 1 with the pathway associated to each gene presented in the heatmap.

2. Checkpoint blockades have demonstrated impressive efficacy in only subsets of patients with a pre-existing T cell immunity⁶, with the response rate is even lower in ovarian cancer even lower

Authors’ response:

This typo mistake is now corrected in the revised version of the manuscript (page 20): “Checkpoint blockades have demonstrated impressive efficacy only in subsets of patients with a pre-existing T cell immunity⁶, with the response rate even lower in ovarian cancer⁴⁰.”

3. Figure 5a, caption: The graph depicts an anti-correlation between the HLA-A gene expression (Log₂(RPKM+1)) and his promoter methylation (beta value) for 48 ovarian cancer cell lines.

Authors’ response:

We’ve corrected Figure 5a caption to the following (page 43):

“The graph depicts an anti-correlation between the HLA-A gene expression (Log₂(RPKM+1)) and its promoter methylation (beta value) for 48 ovarian cancer cell lines.

Desbois, Udyavar, Ryner et al.

Reviewer #2 (Ovarian cancer, TGFb):

This is a detailed report which looks at tissue from ICON 7, and systematically evaluates transcriptomic analysis to investigate mechanisms of T cell exclusion, and attempts to explain impact of this on outcome. It further investigates the biologic basis of effect on different ovarian cancer cell lines, with detailed mechanistic correlation showing effect of TGFb orchestrating T cell exclusion through effect on micro-environment, ECM and fibroblasts, with reversal using TGFb inhibitors. Genetic alterations thought not to be major driver of T cell exclusion but downregulation of MHC-I linked to epigenetic changes - demethylation in tumour cells, with restoration of MHC-I using epigenetic modulators or IFN. TGFb decreases MHC-I and levels of MHC-I restored with TGFb inhibitors. The study also links tumour immunophenotype using a gene expression classifier. These findings portray a potentially elegant explanation for immune exclusion in the microenvironment and the resultant effects on response to therapy and outcome.

There are some key considerations that need some clarification:

Authors' response:

We appreciate the reviewer's comments and feedback. Please find below our point-by-point responses to each of the reviewer's comments:

1. The pathologic selection from ICON7 and TCGA relate to high grade serous subtype, and authors confirmed this selection. However, some of the cell lines selected for further evaluation do not seem to be restricted to serous lines, and so would like the authors to comment if the inferences drawn still apply.

PA-1: ? Teratocarcinoma

TOV112D - ? Endometrioid

COV434 – granulosa

However, SMARCA4 mutations are a feature of small cell ovarian of hypercalcemic type - and dual SMARCA4 and A2 mutations define SCCOHT cell lines -some of the ones cited in this paper have in published reports been thought to have been misclassified. It is essential that the conclusions are based on work done in HGS lines only. The epigenetic work and inhibition by EZH2 should be re-evaluated.

Desbois, Udyavar, Ryner et al.

2. Some detail about the ICON7 patient demographics/treatment outcome for context would be helpful. Please confirm if the specimens were from only control patients?

Authors' response:

We thank the reviewer for the thoughtful comments and constructive suggestions. Please allow us to first make further clarification regarding the second comment. We believe this clarification will help us to better address the reviewer's first comment regarding our choice of cell lines.

1). We apologize for the inaccurate statement in our original manuscript that the three hundred seventy treatment naïve patient samples collected from the phase III ICON7 clinical trial were all **high-grade serous carcinomas**. Following the reviewer's great suggestion, in the revised manuscript, we added a table with demographic and treatment information from the ICON7 study to the supplemental information (Fig. S8). As shown in the Fig. S8, it is important to clarify that the ICON7 cohort we used for classifying and characterizing tumour-immune phenotypes, in fact, included all histotypes of epithelial ovarian tumours (71% serous, 10% clear cell, 9% endometrioid, and 2% mucinous). Although another validation dataset (ovarian cancer TCGA cohort) used in the study only included high-grade serous ovarian patients (Cancer Genome Atlas Research, 2011), it was not our intention to restrict our immune phenotyping study on high-grade serous histology of ovarian cancer.

In the revised manuscript, we also corrected our previous description on histology in ICON7 cohort in the Online Methods section (page 22)

“Three hundred seventy treatment naïve patient samples with **High Grade Serous Carcinoma (HGSC)** were collected from the phase III ICON7 clinical trial” was changed to the following:

“Three hundred seventy treatment naïve patient samples with **epithelial ovarian cancer from mixed histology** were collected from the phase III ICON7 clinical trial (Figure S8).”

	CARBOPLATIN PLUS PACLITAXEL CHEMOTHERAPY	CARBOPLATIN PLUS PACLITAXEL CHEMOTHERAPY PLUS BEVACIZUMAB
n	182	188
Histology	% (n)	% (n)
Serous	71 (130)	69 (130)
Clear cell	10 (18)	13 (25)
Endometrioid	9 (16)	5 (9)
Mucinous	2 (3)	1 (2)
Serous/ClearCell/Endometroid/Mucinous/Other Mixed	5 (9)	9 (16)
Endometrioid/Clear cell/Mixed	1 (1)	2 (3)
PapillaryCystoadenocarcinoma/Unclassified/Undifferentiated/ Adenocarcinoma	3 (5)	2 (3)
Original cancer	% (n)	% (n)
Ovary (epithelial)	90 (164)	88 (165)
Fallopian tube	2 (4)	3 (5)
Primary peritoneum	5 (10)	7 (14)
Mixed	2 (4)	2 (4)
Age Group	% (n)	% (n)
18-39yr	4 (8)	3 (5)
40-64yr	72 (131)	76 (144)
>=65	23 (43)	21 (39)
FIGO Stage	% (n)	% (n)
Stage I	8 (14)	8 (15)
Stage II	11 (20)	14 (26)
Stage III	72 (131)	68 (127)
Stage IV	9 (17)	11 (20)
Grade	% (n)	% (n)
1	5 (10)	4 (8)
2	16 (30)	16 (31)
3	77 (141)	79 (148)
unknown	0 (1)	1 (1)
Platinum Sensitivity	% (n)	% (n)
Sensitive	49 (89)	62 (117)
Intermediate	19 (35)	20 (37)
Resistant	27 (50)	16 (30)
Refractory	1 (2)	0 (0)
NA	3 (6)	2 (4)
Baseline CA-125 Category	% (n)	% (n)
< 2x ULN	49 (89)	39 (74)
>= 2x ULN	50 (92)	60 (112)
NA	1 (1)	1 (2)

Fig. S8. Demographic table of the ICON7 cohort (n=370) including samples used for training and testing sets.

Desbois, Udyavar, Ryner et al.

As additional information for the reviewer, we also compared the prevalence and distribution of tumour-immune phenotypes and molecular subtypes in serous only or all-histology epithelial ovarian tumours in the ICON7 cohort. No significant differences were observed in the serous-only histotype comparing to all-histology epithelial ovarian tumours (See table below)

SEROUS			
	CARBOPLATIN + PACLITAXEL CHEMOTHERAPY	CARBOPLATIN + PACLITAXEL CHEMOTHERAPY PLUS BEVACIZUMAB	ALL TREATMENTS
n	130	130	260
Immune phenotype	% (n)	% (n)	
Infiltrated	39 (51)	38 (49)	38 (100)
Excluded	25 (33)	35 (45)	30 (78)
Desert	31 (40)	23 (30)	27 (70)
Unclassified	5 (6)	5 (6)	5 (12)
Molecular Subtype			
Immunoreactive	29 (38)	28 (36)	28 (74)
Mesenchymal	22 (29)	31 (40)	27 (69)
Proliferative	30 (39)	26 (34)	28 (73)
Differentiative	18 (24)	15 (20)	17 (44)

ALL HISTOTYPES			
	CARBOPLATIN + PACLITAXEL CHEMOTHERAPY	CARBOPLATIN + PACLITAXEL CHEMOTHERAPY PLUS BEVACIZUMAB	ALL TREATMENTS
n	182	188	370
Immune phenotype	% (n)	% (n)	% (n)
Infiltrated	36 (66)	34 (63)	35 (129)
Excluded	20 (37)	30 (57)	25 (94)
Desert	38 (69)	31 (59)	35 (128)
Unclassified	5 (10)	5 (9)	5 (19)
Molecular Subtype			
Immunoreactive	26 (47)	27 (50)	26 (97)
Mesenchymal	20 (36)	29 (54)	34 (90)
Proliferative	35 (64)	30 (57)	33 (121)
Differentiative	19 (35)	14 (27)	17 (62)

Prevalence of tumour-immune phenotypes and molecular subtypes in serous only or all histotypes in the ICON7 cohort.

2). Now, please allow us to address the reviewer's first comment regarding the ovarian cell line selection. Again, we apologize for the inaccurate description on histology of the ICON7 cohort

Desbois, Udyavar, Ryner et al.

in our original manuscript. As we corrected and further clarified in the previous comment, our study was not aimed to restrict the immune phenotype characterization in high-grade serous histology only, but rather included ovarian tumours with all histology in the ICON7 cohort.

Accordingly, while we did pick several cell lines with known serous histology, we did not intend to restrict our *in vitro* cell line analysis in serous histology only. In fact, the panel of ovarian cell lines used in this study represent a wide range of origins and histology of ovarian cancer. As the reviewer pointed out, accurately identifying each cell line's histotype has shown to be challenging, even though several previous studies have attempted to characterize the histopathological origin of some of ovarian cell lines through genomic comparison with tumour samples (Domcke et al., 2013).

For our experimental settings, we did not focus on the histological origin but rather on the HLA gene expression and methylation score (Figure 5a). We selected cell lines for which we have a low methylation score and high HLA-A gene expression (SK-OV-3 and OVCA-420) or in contrast a high methylation score and low HLA-A gene expression (OAW42 and PA-1) to treat them either with TGFβ or 5-Aza.

Lastly, we agree with the reviewer that the two cell lines with SMARCA4 mutations, COV434 and TOV1120D, and PA-1 are likely not representative of ovarian epithelial carcinoma and therefore indeed not appropriate to use here. In the revised manuscript, we have now removed the results from these cell lines from the supplemental Figure S6. We have also removed the following sentences from the main text (page 14)

~~“Lastly, a previous study has shown that a subset of cancers harbouring mutations in the SWI/SNF ATPase, SMARCA4, is sensitive to EZH2 inhibition³⁴. Indeed, two ovarian cancer cell lines with SMARCA4 mutations, COV434 and TOV112D, showed increased HLA-A expression upon treatment with the EZH2-targeting histone methyltransferase inhibitor, EPZ-6438 (Fig. S5c).”~~

3. Were the ICON7 experimental arm patients with bev evaluated?

Author response:

The short answer for this question is that patient samples in bev arm were evaluated in terms of immune phenotypes and molecular subtypes, but not included in survival association analysis. In more details, among the 370 tumour samples from ICON7 analyzed in this study, 182 were from patients treated with chemotherapies (carboplatin + paclitaxel) control arm and

Desbois, Udyavar, Ryner et al.

188 samples were from patients who received the combination of chemotherapies + Bevacizumab. To generate the molecular classifier and study the molecular determinant characterizing the different tumour-immune phenotypes, we used samples from both arms regardless of the treatments. However, for the survival association with the tumour-immune phenotypes or molecular subtypes (Figure 2f), we only included patients from the chemotherapies arm and excluded the patients from the experimental arm with Bevacizumab. The reason for this was that our purpose of this study was to investigate the prognostic role of the different tumour-immune phenotypes and not the impact of the experimental drug treatment.

4. Is there any information on BRCA status and HRD of the ICON7 samples used and was impact of this assessed

Author response:

We thank the reviewer for this constructive suggestion and have performed additional genetic mutation analysis on subset of ICON7 samples in the revised manuscript. Unfortunately, we do not have enough tissue in our ICON7 collection to perform genome-wide mutation profiling and HRD assessment. Instead, we performed targeted sequencing on 88 oncogenes (including both BRCA1 and BRCA2) in 271 samples using a MMP-seq panel developed in-house (Bourgon et al., 2014). No significant association was overserved between BRCA mutation status and the tumor-immune phenotypes (see Figure S5b).

In the revision, we added the following updates in the main text: “Due to limited tissue availability in ICON7 collection for genome-wide mutation profiling and HRD assessment, we performed targeted sequencing on 88 oncogenes (including both BRCA1 and BRCA2) in 271 samples using a MMP-seq panel (Bourgon et al., 2014). No significant association was overserved between BRCA mutation status and the tumor-immune phenotypes (see Figure S5b).” (page 11)

Desbois, Udyavar, Ryner et al.

Fig. S5b BRCA mutation status (BRCA1 or BRCA2) across the tumour-immune phenotypes in the ICON7 cohort. 216 ovarian cancer samples had both mutation data and characterized immune phenotypes. The graph depicts % of mutation (MUT) or wild type (WT) in each immune phenotype.

5. Were there any m/v analysis done to look at impact of immune phenotype in relation to stage/residual disease on PFS and its potential confounding effect on Fig 2f. Was there any influence of immune phenotype on stage etc.

Author response:

Thank you for the reviewer's excellent suggestions. For the revision of this manuscript, we have now added a multivariate analysis of the association of the tumour-immune phenotype in combination with stage with PFS (Fig. S3, below). We confirmed that patients with late stage disease (stage III and IV) were significantly associated with poor prognosis in the ICON7 cohort. However, the association between excluded immune phenotype and poor prognosis remained significant even after correction of potential effect of the tumour stages (Fig. S3).

In the revised manuscript, we have added a supplemental Figure S3 on the multivariate analysis and updated the main text as following (page 8)

"Finally, we performed multivariate analysis to include tumour stage, a known prognosis factor in ovarian cancer. We confirmed that patients with late stage disease (stage III and IV) were significantly associated with poor prognosis in the ICON7 cohort. However, the association between excluded immune phenotype and poor prognosis remained significant even after correction of potential effect of the tumour stages (Fig. S3)."

Fig. S3 Association of the tumour-immune phenotypes with PFS in a multivariate analysis correcting for the tumour stage in the ICON7 cohort (control chemo arm only).

Desbois, Udyavar, Ryner et al.

Reviewer #3 (Ovarian cancer, pathology):

1. Based on CD8+ T cell quantity and epithelial distribution ovarian cancer cases were classified into infiltrated, excluded and desert. The gold standard for this categorization is unclear. The associations of these categories with survival (Fig. 2f) make no sense: it has been consistently shown that the presence and quantity of T cell infiltration is positively associated with survival e.g. PMID: 29049607. Hence, the fact that infiltrated and deserted tumors (Fig 2f) have the same outcome undermines the credibility of the proposed classification, which is the basis of this manuscript.

Author response:

We appreciate the reviewer's comments and would like to take this opportunity to make further clarification on these important points.

As the reviewer pointed out, lacking a gold standard of immune phenotype classification was indeed a great challenge we and others have been facing. As we introduced in the introduction part of the manuscript, although different immune infiltration patterns, i.e. infiltrated (or inflamed), excluded and desert have been previously described in other solid cancers (Hegde et al., 2016), it remained challenging to systematically define the tumour-immune phenotype of most cancer patients using the traditional pathologist review method, largely due to the highly heterogeneous and complex nature of immune cell infiltration and distribution patterns in the tumour microenvironment.

On the other hand, it was also because of this challenge that motivated us to take a novel approach in this study and develop a more quantitative metric based on digital pathology to better defined these immune phenotypes. Our approach confirmed that the infiltration pattern of the CD8+ T cells was in fact a continuum rather than discrete categories (Figure 1 and 2d). Instead of arbitrarily imposing a cutoff which may be completely biologically or clinically-irrelevant, we took a novel approach by integrating the digital pathology and transcriptome analysis and turned a histological metric into a molecular metric for defining immune phenotypes. In another word, we let the molecular features to better define the immune phenotypes. More importantly, this approach also allowed us to further characterize the biology and key pathways underlying these immune phenotypes.

Regarding the association of immune phenotypes with progression-free survival in the ICON7 chemo cohort, we also appreciated the reviewer's comments. Indeed, several studies have shown that the numbers of tumour infiltrating CD8⁺ T cells were associated with good prognosis in ovarian cancer (Hwang et al., 2012; Ovarian Tumor Tissue Analysis et al., 2017; Zhang et al., 2003), we were also initially surprised by the finding that the infiltrated and desert

Desbois, Udyavar, Ryner et al.

phenotypes both showed similar outcomes in the ICON7 cohort (Fig. 2f). However, after carefully looking into the data, we believe there might be several factors contributed to this observation:

1. First, our data suggested that it might be an over-simplification to conclude that TIL+ and TIL- tumour classification trumps other biological factors in determining distinct clinical outcomes. As discussed in the results (page 8) and discussion (page 18) sections in our manuscript, we observed that the immune desert tumours in ovarian cancer had heterogeneous underlying biology and were comprise of two distinct molecular subtypes, the differentiated and the proliferative subtype. These two molecular subtypes showed different clinical outcomes in previous ovarian cancer study (Tothill et al., 2008) and in the ICON7 cohort of this study (Fig. 2f). We think these findings underline the importance of studying tumour-immune phenotypes in the context of disease biology.
2. In a separate study, we also have similar observations in HCC (unpublished data), where immune desert tumours also consist of two molecular subtypes with different disease biology, i.e. the stem-like subtype vs. the differentiated subtype with CTNNB1 mutations. Incidentally, the stem-like subtype is associated with the poorest prognosis in HCC, while the differentiated subtype shows the best prognosis. Therefore, using TIL+ vs. TIL- as a single metric to predict prognosis may be confounded by different underlying biology.
3. Looking into more nuanced details, we believe that different methods and criteria for assessing and interpreting immune infiltration patterns may also contribute to the apparent discrepancy between the published results and our study. For example, most of the published studies (Hwang et al., 2012; Lo et al., 2017; Ovarian Tumor Tissue Analysis et al., 2017; Zhang et al., 2003) focused on the infiltration of T cells in the tumour epithelium (equivalent to our infiltrated immune phenotype), but possibly lumped together the rest of tumours with T cells in the surrounding stroma (equivalent of our excluded immune phenotype) and tumours with little T cells (equivalent of desert tumours). Indeed, as shown in Fig. S7, we observed similar results as previously reported when we combined the desert and excluded tumours (low CD8+ T cell in the tumour epithelium) in opposition to infiltrated tumours (high CD8+ T cell infiltration in the tumour epithelium). While we did not get significance here, we observed a trend for worse survival for patients having desert or excluded tumours comparing to infiltrated tumours, similar to the study cited by the reviewer

Fig. S7. Association of the progression free survival with the tumour-immune phenotype, comparing infiltrated or excluded/desert tumours. The graph includes patients in the chemo control arm for the ICON7 cohort (n=172).

In the revised manuscript, we have added Fig. S7 and the following statements to further clarify these points:

1. In the introduction in main text, we added the follow sentences to provide references of previous studies (page 3):

“Several studies have also shown that the numbers of tumour infiltrating CD8⁺ T cells were associated with good prognosis in ovarian cancer (Hwang et al., 2012; Ovarian Tumor Tissue Analysis et al., 2017; Zhang et al., 2003).”

2. In the discussion in the main text (page 17), we added the following discussion on the survival association analysis:

“As we introduced earlier, several studies have reported that the numbers of tumour infiltrating T cells were associated with good prognosis in ovarian cancer¹⁰⁻¹². However, most of these studies focused on the infiltration of T cells in the tumour epithelium (equivalent to our infiltrated immune phenotype), but possibly lumped together the rest of tumours with T cells in the surrounding stroma (equivalent of our excluded immune phenotype) and tumours with little T cells (equivalent of desert tumours). Indeed, as shown in Fig. S7, we observed similar

Desbois, Udyavar, Ryner et al.

results as previously reported when we combined the desert and excluded tumours (low CD8+ T cell in the tumour epithelium) in opposition to infiltrated tumours (high CD8+ T cell infiltration in the tumour epithelium). We believe our study added an important additional layer of information, i.e. spatial distribution of T cells comparing to the previous studies and identified that the excluded phenotype was associated with the worst prognosis in epithelial ovarian cancer. In addition, one of the key conclusions we drew from this study was that tumour-immune phenotypes should be studied and interpreted in the context of disease biology. For example, we observed that the immune desert tumours in ovarian cancer are heterogeneous and comprise of two distinct molecular subtypes, the differentiated and the proliferative subtype. These two molecular subtypes showed different clinical outcomes in previous ovarian cancer study (Tothill et al., 2008) and in the ICON7 cohort of this study (Fig. 2f). Our findings call for more attention on understanding the underlying disease biology to avoid potential oversimplified classification of TIL+ and TIL- tumours with distinct biology lumped together.”

2. Because of limited access to CD8 IHC slides, the rationale of developing a gene expression classifier is stated. My question would be: is it not easier to fix the logistics and get CD8 IHC slides? What is the accuracy (and sensitivity/specificity) of the gene expression surrogate (because if it is not >95% it may not be sufficient in the era of precision medicine)?

Author response:

As discussed in the previous comment, what motivated us to turn a histologic metric into a molecular metric for defining immune phenotypes was not because of limited access to CD8 IHC slides, but due to the observed continuum nature of immune infiltration pattern quantified by digital pathology. Whether our molecular classifier for immune phenotypes could be further developed into a diagnostic tool to guide patient treatment will need to be further evaluated. On the other hand, we believe the most valuable contribution of our study was that our novel approach enabled using immune phenotypes as a useful framework to characterize and better understand the molecular features and underlying biology associated with these immune phenotypes. These learning in turn may inform novel therapeutic strategies to overcome immune suppression and optimize the efficacy of immunotherapies.

3. The quality of the pathology review of ICON7 is questionable: Fig 3F shows a relevant number of cases with dMMR, which are endometrioid carcinoma and NOT high-grade serous. This mix will confound analysis, e.g. high-grade serous molecular subtypes were assigned to endometrioid carcinomas etc.

Author response:

We thank the reviewer for the comments and have revised our manuscript to clarify the demographic and histological characteristics of the ICON7 cohort. As the newly added demographic table clarified (Fig. S8), the ICON7 cohort we used for classifying and characterizing tumour-immune phenotypes in fact included all histotypes of epithelial ovarian tumours (71% serous, 10% clear cell, 9% endometrioid, and 2% mucinous). Although another validation dataset (ovarian cancer TCGA cohort) used in the study only included high-grade serous ovarian patients (Cancer Genome Atlas Research, 2011), it was not our intention to restrict our immune phenotyping study on high-grade serous histology of epithelial ovarian cancer.

	CARBOPLATIN PLUS PACLITAXEL CHEMOTHERAPY	CARBOPLATIN PLUS PACLITAXEL CHEMOTHERAPY PLUS BEVACIZUMAB
n	182	188
Histology	% (n)	% (n)
Serous	71 (130)	69 (130)
Clear cell	10 (18)	13 (25)
Endometrioid	9 (16)	5 (9)
Mucinous	2 (3)	1 (2)
Serous/ClearCell/Endometroid/Mucinous/Other Mixed	5 (9)	9 (16)
Endometrioid/Clear cell/Mixed	1 (1)	2 (3)
PapillaryCystoadenocarcinoma/Unclassified/Undifferentiated/ Adenocarcinoma	3 (5)	2 (3)
Original cancer	% (n)	% (n)
Ovary (epithelial)	90 (164)	88 (165)
Fallopian tube	2 (4)	3 (5)
Primary peritoneum	5 (10)	7 (14)
Mixed	2 (4)	2 (4)
Age Group	% (n)	% (n)
18-39yr	4 (8)	3 (5)
40-64yr	72 (131)	76 (144)
>=65	23 (43)	21 (39)
FIGO Stage	% (n)	% (n)
Stage I	8 (14)	8 (15)
Stage II	11 (20)	14 (26)
Stage III	72 (131)	68 (127)
Stage IV	9 (17)	11 (20)
Grade	% (n)	% (n)
1	5 (10)	4 (8)
2	16 (30)	16 (31)
3	77 (141)	79 (148)
unknown	0 (1)	1 (1)
Platinum Sensitivity	% (n)	% (n)
Sensitive	49 (89)	62 (117)
Intermediate	19 (35)	20 (37)
Resistant	27 (50)	16 (30)
Refractory	1 (2)	0 (0)
NA	3 (6)	2 (4)
Baseline CA-125 Category	% (n)	% (n)
< 2x ULN	49 (89)	39 (74)
>= 2x ULN	50 (92)	60 (112)
NA	1 (1)	1 (2)

Fig. S8. Demographic table of the ICON7 cohort (n=370) including samples used for training and testing sets.

As additional information for the reviewer, we also compared the prevalence and distribution of tumour-immune phenotypes and molecular subtypes in serous only and all histology ovarian tumours in the ICON7 cohort and observed no significant differences (See figure below)

SEROUS			
	CARBOPLATIN + PACLITAXEL CHEMOTHERAPY	CARBOPLATIN + PACLITAXEL CHEMOTHERAPY PLUS BEVACIZUMAB	ALL TREATMENTS
n	130	130	260
Immune phenotype	% (n)	% (n)	
Infiltrated	39 (51)	38 (49)	38 (100)
Excluded	25 (33)	35 (45)	30 (78)
Desert	31 (40)	23 (30)	27 (70)
Unclassified	5 (6)	5 (6)	5 (12)
Molecular Subtype			
Immunoreactive	29 (38)	28 (36)	28 (74)
Mesenchymal	22 (29)	31 (40)	27 (69)
Proliferative	30 (39)	26 (34)	28 (73)
Differentiative	18 (24)	15 (20)	17 (44)
ALL HISTOTYPES			
	CARBOPLATIN + PACLITAXEL CHEMOTHERAPY	CARBOPLATIN + PACLITAXEL CHEMOTHERAPY PLUS BEVACIZUMAB	ALL TREATMENTS
n	182	188	370
Immune phenotype	% (n)	% (n)	% (n)
Infiltrated	36 (66)	34 (63)	35 (129)
Excluded	20 (37)	30 (57)	25 (94)
Desert	38 (69)	31 (59)	35 (128)
Unclassified	5 (10)	5 (9)	5 (19)
Molecular Subtype			
Immunoreactive	26 (47)	27 (50)	26 (97)
Mesenchymal	20 (36)	29 (54)	34 (90)
Proliferative	35 (64)	30 (57)	33 (121)
Differentiative	19 (35)	14 (27)	17 (62)

Prevalence of tumour-immune phenotypes and molecular subtypes in serous only or all histotypes of epithelial ovarian cancer in the ICON7 cohort.

Desbois, Udyavar, Ryner et al.

4. The studied cell lines were not carefully chosen: e.g. OAW42 and SKOV3 are unlikely high-grade serous (PMID: 23839242).

Author response:

As we have clarified in the response to the previous comment, the ICON7 cohort used in this study comprised of 70% of high-grade serous and 30% of mixed histology of epithelial ovarian cancer. It was not our intention to restrict our immune phenotyping study on high-grade serous histology of epithelial ovarian cancer, we therefore did not try to restrict our cell line *in vitro* studies to serous histology either, especially characterization of the histological origins of ovarian cell lines have been shown to be challenging.

5. Relevant ovarian cancer literature is not cited, e.g. TGFb in not novel, there are already reviews, e.g. 31091744. No citation of any CD8 study e.g. PMID: 29049607

Author response:

We thank the reviewer for the constructive suggestions. In the revised manuscript, we added additional literature and discussion on the role of TGFb in ovarian cancer and included studies that investigated the infiltration of CD8+ T cells in this cancer (page 20):

Page 20, “Additionally, many previous studies have reported a direct immune suppressive role of TGFβ on T cells, NK cells and dendritic cells and promoting T regulatory cell differentiation^{42,43}. Altogether, our study provided additional evidence supporting a complex and multi-faceted role of TGFβ in immune modulation in cancer.”

Page 21, ‘We have also demonstrated previously that in primary and metastatic mouse colorectal cancer models, blocking TGFβ and PD-L1 signaling pathways triggered a strong T cell infiltration in the tumour core and enhanced tumour regressions and survival^{8,46} supporting additional studies in breast and colorectal mouse cancer models^{46,47}.’”

6. Line 79 CD8 IHC on N=370 from ICON7 versus line 124 CD8 IHC data for 122 out of the 215 samples?

Author response:

We apologize for the misleading statement here and would like to take the opportunity to clarify these numbers. Our dataset collection for ICON7 accounts for 370 samples. All 370 samples had CD8 IHC and RNA sequencing data. However, only 277 out of the 370 samples had

Desbois, Udyavar, Ryner et al.

digital pathology done on the CD8 IHC slides. To build our molecular classifier, we bagged the samples into 2 sets: i) a training set (n=155) and ii) a testing set (n=215). All samples from the training set had both digital pathology analysis done on the CD8 IHC and RNA sequencing data. However, among the samples used for the testing set only 122 out of the 215 samples had digital pathology analysis. Hence, only 277 samples had a digital pathology score. We believe this do not affect our analysis. The digital pathology score for the 122 samples from the testing set served only to the visual confirmation as depicted in Figure 2d.

In the revised manuscript, we updated our text in the main result section as follow:

Page 6, “In this approach, transcriptome analysis was integrated with the digital pathology analysis on the same set of samples from the ICON7 trial. Using a training set of 155 samples for which we have digital pathology scores, we identified 352 genes whose expression can be predicted by the quantity (R) and/or spatial distribution of CD8+ T cells (θ) using a random forest regression model (See Methods, Fig. S1a-b, Table S1).”

Page 7, “CD8 IHC data and digital pathology analysis were available only for 122 out of the 215 tumour samples from the testing set.”

Page 23, In the method section: “Digital pathology analysis was performed on 277 out of 370 tumours from the ICON7 collection (155 out of 155 samples from the training set and 122 out of 215 samples from the testing set had digital pathology scores) and the whole vendor procured collection.”

7. The lack of consistency between "primary" putative ovarian site and "metastases" putative peritoneal site is concerning (Fig. S4B). Depending whether a tumor sample is taken from the ovary versus peritoneum, the frequency of 'excluded' will increase from 13% to 62%. The results need to be controlled for the sampling site.

Author response:

This is an interesting point raised by the reviewer. We would like to clarify that the vast majority of the training, testing and validation datasets used in this study, including ICON7 cohort and TCGA cohort, were derived from primary ovarian tumours (Fig. S8). Hence, the prevalence of tumour-immune phenotypes reported in this study has been consistent and primarily representative of primary ovarian tumours. While in one of the small cohorts used in *in situ* validation (n = 89), we did include a small number of metastatic tumours (depicted in the original Figure S4, now Figure S5), we observed higher prevalence in excluded tumours in the

Desbois, Udyavar, Ryner et al.

metastatic tumours comparing to primary tumours. Although this finding could indicate potential evolution of tumour-immune phenotypes post-chemotherapy or along disease progression (all met samples were collected at disease progression post chemotherapy), further validation in larger dataset of paired primary vs. metastatic tumours as well as treatment-naïve vs. post chemotherapy samples is needed to confirm these findings.

In the revised manuscript, we further clarified these points by adding the following changes:

Online methods, page 22:

“An independent validation collection (n=84) was procured from Cureline, Inc (Brisbane, CA, US). These include 55 treatment-naïve primary tumours and 29 paired metastatic tumours collected at disease progression post front line chemotherapy.”

Results, page 12:

“To validate these findings and distinguish which cell compartment underwent these molecular changes, we performed in situ analysis on an independent ovarian tumour collection consisting of 84 ovarian tumour samples procured from a vendor. RNAseq transcriptome analysis was performed on these samples and their tumour-immune phenotypes were predicted based on the 157-gene classifier developed in this study (Fig. S5c-d). Interestingly, we observed a much higher prevalence of excluded phenotype in the metastatic tumours comparing to the primary tumours (Fig. S5c), suggesting immune phenotypes may evolve post-chemotherapy or along disease progression. Further validation in larger dataset of paired primary vs. metastatic tumours as well as treatment-naïve vs. post chemotherapy samples is needed to confirm these findings”

Desbois, Udyavar, Ryner et al.

References

- Bourgon, R., Lu, S., Yan, Y., Lackner, M. R., Wang, W., Weigman, V., Wang, D., Guan, Y., Ryner, L., Koeppen, H., *et al.* (2014). High-throughput detection of clinically relevant mutations in archived tumor samples by multiplexed PCR and next-generation sequencing. *Clinical cancer research : an official journal of the American Association for Cancer Research* 20, 2080-2091.
- Campoli, M., and Ferrone, S. (2008). HLA antigen changes in malignant cells: epigenetic mechanisms and biologic significance. *Oncogene* 27, 5869-5885.
- Cancer Genome Atlas Research, N. (2011). Integrated genomic analyses of ovarian carcinoma. *Nature* 474, 609-615.
- de Jong, R. A., Boerma, A., Boezen, H. M., Mourits, M. J., Hollema, H., and Nijman, H. W. (2012). Loss of HLA class I and mismatch repair protein expression in sporadic endometrioid endometrial carcinomas. *International journal of cancer Journal international du cancer* 131, 1828-1836.
- Domcke, S., Sinha, R., Levine, D. A., Sander, C., and Schultz, N. (2013). Evaluating cell lines as tumour models by comparison of genomic profiles. *Nature communications* 4, 2126.
- Hegde, P. S., Karanikas, V., and Evers, S. (2016). The Where, the When, and the How of Immune Monitoring for Cancer Immunotherapies in the Era of Checkpoint Inhibition. *Clinical cancer research : an official journal of the American Association for Cancer Research* 22, 1865-1874.
- Hicklin, D. J., Marincola, F. M., and Ferrone, S. (1999). HLA class I antigen downregulation in human cancers: T-cell immunotherapy revives an old story. *Mol Med Today* 5, 178-186.
- Hiraki, A., Fujii, N., Murakami, T., Kiura, K., Aoe, K., Yamane, H., Masuda, K., Maeda, T., Sugi, K., Darzynkiewicz, Z., *et al.* (2004). High frequency of allele-specific down-regulation of HLA class I expression in lung cancer cell lines. *Anticancer Res* 24, 1525-1528.
- Hwang, W. T., Adams, S. F., Tahirovic, E., Hagemann, I. S., and Coukos, G. (2012). Prognostic significance of tumor-infiltrating T cells in ovarian cancer: a meta-analysis. *Gynecologic oncology* 124, 192-198.
- Lo, C. S., Sanii, S., Kroeger, D. R., Milne, K., Talhouk, A., Chiu, D. S., Rahimi, K., Shaw, P. A., Clarke, B. A., and Nelson, B. H. (2017). Neoadjuvant Chemotherapy of Ovarian Cancer Results in Three Patterns of Tumor-Infiltrating Lymphocyte Response with Distinct Implications for Immunotherapy. *Clinical cancer research : an official journal of the American Association for Cancer Research* 23, 925-934.

Desbois, Udyavar, Ryner et al.

Mehta, A. M., Jordanova, E. S., Kenter, G. G., Ferrone, S., and Fleuren, G. J. (2008). Association of antigen processing machinery and HLA class I defects with clinicopathological outcome in cervical carcinoma. *Cancer immunology, immunotherapy* : CII 57, 197-206.

Ovarian Tumor Tissue Analysis, C., Goode, E. L., Block, M. S., Kalli, K. R., Vierkant, R. A., Chen, W., Fogarty, Z. C., Gentry-Maharaj, A., Toloczko, A., Hein, A., *et al.* (2017). Dose-Response Association of CD8+ Tumor-Infiltrating Lymphocytes and Survival Time in High-Grade Serous Ovarian Cancer. *JAMA Oncol* 3, e173290.

Ritter, C., Fan, K., Paschen, A., Reker Hardrup, S., Ferrone, S., Nghiem, P., Ugurel, S., Schrama, D., and Becker, J. C. (2017). Epigenetic priming restores the HLA class-I antigen processing machinery expression in Merkel cell carcinoma. *Scientific reports* 7, 2290.

Tothill, R. W., Tinker, A. V., George, J., Brown, R., Fox, S. B., Lade, S., Johnson, D. S., Trivett, M. K., Etemadmoghadam, D., Locandro, B., *et al.* (2008). Novel molecular subtypes of serous and endometrioid ovarian cancer linked to clinical outcome. *Clinical cancer research : an official journal of the American Association for Cancer Research* 14, 5198-5208.

Wallin, J. J., Bendell, J. C., Funke, R., Sznol, M., Korski, K., Jones, S., Hernandez, G., Mier, J., He, X., Hodi, F. S., *et al.* (2016). Atezolizumab in combination with bevacizumab enhances antigen-specific T-cell migration in metastatic renal cell carcinoma. *Nature communications* 7, 12624.

Zhang, L., Conejo-Garcia, J. R., Katsaros, D., Gimotty, P. A., Massobrio, M., Regnani, G., Makrigiannakis, A., Gray, H., Schlienger, K., Liebman, M. N., *et al.* (2003). Intratumoral T cells, recurrence, and survival in epithelial ovarian cancer. *The New England journal of medicine* 348, 203-213.

REVIEWER COMMENTS

Reviewer #1 (Remarks to the Author):

The authors have addressed all concerns raised satisfactorily. Hence, we recommend accept.

Reviewer #4 (Remarks to the Author):

This reviewer is reading the manuscript for the first time – I have been asked specifically to comment on whether the authors have adequately addressed the comments from the reviewers two and three.

Overall, I think that this is high quality work that address an important clinical question.

Reviewer two raised several points

1. High grade serous carcinoma pathology vs other histological types

ICON7 enrolled patients of all histological types as well as all stages (although the number of patients with stage I - IIA disease was restricted to 10% total recruitment). I agree that Table S8 is a useful addition to the manuscript.

However, it is important to note that the numbers of non-high grade serous pathology mean that it is difficult fully to justify the authors' comment in the rebuttal "it was not our intention to restrict our immune phenotyping study on high-grade serous histology of ovarian cancer". 260/371 tumours were HGSC and thus the study is underpowered to make inferences about the specifics of immune cell infiltration in other histologies. As detailed in the manuscript, the authors indicate that there is no obvious link between immune phenotype (either IHC or gene expression) and BRCA1/2 mutation status or TMB in the TCGA dataset (Fig 3f). However, it remains likely that there are tumour-autonomous genomic/epigenomic drivers of immune infiltration and gene expression, which will differ between histological types of ovarian cancer. Thus, some caution is required in making broad general comments when 70% data derive from one histological subtype.

2. The nature of the cell lines.

There is robust and on-going debate as to the utility of many ovarian cancer cell lines (especially SKOV3) since the publication of Domcke et al in Nature Communications in 2013. I think that the authors have used a broad panel of ovarian cancer cells and justify their selection adequately. Clearly, there were experiments on EZH2 inhibition in the original manuscript that have been removed as well as discussion on the role of SMARCA4 – I agree with reviewer 2 that these are probably only tangentially related to most ovarian cancer subtypes.

3. The control vs experimental arm patients from ICON7

The authors have made the logical decision to include the experimental arm tumours in the phenotyping analysis but not in the survival associations – this makes complete sense. However, given what is known about role of VEGF signalling in immune responses, it would be intriguing to see whether bevacizumab treatment influences outcomes in the defined immune subsets. Data were presented at ASCO in 2014 (but not published as far as I am aware) to suggest that there were gene expression subsets that did not derive benefit from the addition of bevacizumab in ovarian cancer.

4. HRD status/BRCA status in the ICON7 samples

The authors have analysed 88 genes, including BRCA1/2, in the ICON7 samples, and show no correlation immune phenotype (INF vs EC vs DES). This is an important addition to the manuscript.

5. Multivariate analysis

This is incredibly important, and the authors have included stage in a multivariate analysis.

However, two additional critical clinical factors in any survival analysis in ovarian cancer are age and debulking status. There is widespread evidence that age influences immune responses in all diseases, so I think that age would be an important co-variate in these analyses. ICON7 also collected information on debulking status, and the the ICON7 high risk subset of patients included those with residual disease. Moreover, the original publication from George Coukos (Zhang et al 2003) showed that the only group with improved outcomes by debulking status (optimal vs sub-optimal debulking) were those intra-tumoural T cells, implying a link between immune infiltration and debulkability (see Figure 2F in Zhang et al). Thus, it will be important to include associations between immune subsets and debulking status in ICON7, even as a descriptive analysis.

Reviewer three made several comments

1. Gold standard for defining INF vs EXC vs DES

The reviewer raised an extremely important point – how were tumours allocated into the three IHC groups. I acknowledge the authors' reply that figure 1b shows a continuum of immune cell infiltration states (which is physiologically logical). I note that the classifiers of INF vs EXC vs DES were performed using Definiens software, but the online methods section is not clear on how was this developed nor what manual validation/scoring was performed by a certified pathologist (or even if any manual validation was performed). It would be helpful if the classification parameters that the authors developed were made available so that other investigators could attempt to recreate this work using the same parameters on other sample sets.

I note Figure S7, which is an analysis of INF vs EXC+DES, which is an attempt to recreate other analysis, including the OTTA analysis of 2017, and potentially reinforces the challenges of binary classifiers (TIL vs no TIL).

Overall, I think that the authors have addressed the reviewer comments clearly here.

2. Is CD8 IHC still not the optimum assay for defining immune status rather than adding the complication of gene expression.

The reviewer raises the important point that every pathology lab in every hospital will be able to undertake CD8 IHC, whilst clinical-grade gene expression analyses are much more challenging to implement. I would also raise the fact that the authors here have developed their own gene expression signature set for their primary analyses, with previous signatures (e.g. Tothill, TCGA) as secondary analyses. A very large OTTA/OCAC gene expression signature for HGSC has just been published online (PMID 32473302) that would also be relevant to discuss as it will be important (not least for management of patients) that an agreed signature be developed, validated and applied prospectively from now on.

3. Histopathology of ICON7 and dMMR

The authors point out that there were some endometrioid cases in ICON7, so the presence of dMMR is explicable.

I would also point out that the authors mis-spell endometrioid in Table S8 – it is endometrioid NOT endometroid.

4. Cell lines

See reply to reviewer 2.

5. Whether appropriate literature has been cited

The authors address this point satisfactorily.

6. Numbers of samples analysed

Again, the authors address this point satisfactorily.

7. Primary site vs metastasis

The reviewer raises an extremely important point here that I think the authors have slightly misunderstood. The authors now present data on recurrent disease – the rebuttal quotes the online methods: “An independent validation collection (n=84) was procured from Cureline, Inc (Brisbane, CA, US). These include 55 treatment-naïve primary tumours and 29 paired metastatic tumours collected at disease progression post front line chemotherapy”. However, this is recurrent disease rather than metastatic disease: 70% women with ovarian cancer already have metastatic disease at the time of diagnosis (Stage III or IV disease). The question of how primary tumours may differ from disease that has spread to the omentum and other sites in the peritoneal cavity at the time of diagnosis is vital for the management of newly-diagnosed patients. More importantly, 50% or more of women with advanced disease are now treated with primary/neoadjuvant chemotherapy following a biopsy, almost invariably from the omentum or peritoneal disease and so the only disease that will be assessed will be metastatic. Moreover, there are strong published data to show that primary chemotherapy alters immune cell infiltration (see e.g. PMID 27306793), meaning that assessment on chemotherapy-naïve tissues are required. Thus, for true clinical utility, the classifiers need to be evaluated on a large number of metastatic samples rather than primary tumours only and there needs to be a clear statement in the discussion about potential discrepancies between primary tumour and disease in e.g. the omentum: the work of Sohrab Shah and Brad Nelson (Zhang et al (2018) Cell 173:1755 and also the recently published work from Martin Miller (Jimenez-Sanchez et al Nature Gen. (2020) 52:582 are important here.

Additional minor comments

1. Line 129: what does ‘sensible’ mean in this context?
2. Line 219: I think that the 10th word in this line should be ‘observed’ rather than ‘over served’.
3. Line 229 – 30: the term ‘genetic alterations’ is a little broad. The authors have look at mutational burden (from a very small panel of genes), dMMR and HR status. That is a narrow spectrum of alterations, especially in light of the fact that a) only 88 genes were sequenced and b) high grade serous carcinoma is driven by copy number alterations and structural variants.
4. Line 296: hypermethylation is mis-spelled.
5. Line 431: “this study provided the first systematic and in-depth characterization of the molecular features and mechanisms underlying the tumour-immune phenotypes in human cancer.” I think that this statement is perhaps hyperbole – see Zhang et al (2018) Cell 173:1755 and also Jimenez-Sanchez et al Nature Gen. (2020) 52:582 as two examples in ovarian cancer alone.

Responses to Reviewers' Comments

Reviewer #1 (Remarks to the Author):

The authors have addressed all concerns raised satisfactorily. Hence, we recommend accept.

Authors' response:

We greatly appreciate the reviewer's recommendation and feedback. We would like to thank her/him again for the careful review of our work and constructive comments, which contributed to the further improvement of our manuscript.

Reviewer #4 (Remarks to the Author):

This reviewer is reading the manuscript for the first time – I have been asked specifically to comment on whether the authors have adequately addressed the comments from the reviewers two and three.

Overall, I think that this is high quality work that address an important clinical question.

Authors' response:

We greatly appreciate the reviewer's overall positive opinions on our work. We would like to thank the reviewer for carefully assessing our responses to the comments from reviewers #2 and #3, and providing constructive comments for further improving the clarity of our manuscript. Below we have addressed the reviewer's comments point by point. We have also made corresponding changes/additions in the revised manuscript to further improve the clarity of our work.

1. Reviewer two raised several points

1. High grade serous carcinoma pathology vs other histological types

ICON7 enrolled patients of all histological types as well as all stages (although the number of patients with stage I - IIA disease was restricted to 10% total recruitment). I agree that Table S8 is a useful addition to the manuscript.

However, it is important to note that the numbers of non-high grade serous pathology mean that it is difficult fully to justify the authors' comment in the rebuttal "it was not our intention to restrict our immune phenotyping study on high-grade serous histology of ovarian cancer". 260/371 tumours were HGSC and thus the study is underpowered to make inferences about the specifics of immune cell infiltration in other histologies. As detailed in the manuscript, the authors indicate that there is no obvious link between immune phenotype (either IHC or gene

expression) and BRCA1/2 mutation status or TMB in the TCGA dataset (Fig 3f). However, it remains likely that there are tumour-autonomous genomic/epigenomic drivers of immune infiltration and gene expression, which will differ between histological types of ovarian cancer. Thus, some caution is required in making broad general comments when 70% data derive from one histological subtype.

Authors' response:

We agree with the reviewer that although we did not restrict our immune phenotyping study on high-grade serous histology of ovarian cancer, our current study was underpowered to draw firm conclusions on other histology given their relative lower representations in our study. In the revised manuscript, we included the following changes to the results and discussion sections to further clarify these points:

Page 11: 'Based on the RNAseq data, we first predicted the tumour-immune phenotypes for 412 high-grade serous (HGS) ovarian tumour samples in the TCGA dataset by applying our 157-gene molecular classifier (Fig. S5a and Table S2). Our analysis revealed an overall absence of significant association between tumour-immune phenotypes and TMB, neo-antigen load, dMMR or homologous recombination deficiency (HRD) in HGS ovarian cancer, with an exception that a slightly lower neoantigen load was observed in the desert compared to the infiltrated tumours (Fig. 3f). Together, our results suggest that these genetic alterations may not be a major driver of the infiltration or exclusion of CD8+ T cells in HGS ovarian cancer.'

Page 21: 'It is also worth to note some of the limitations of this study. First, although we did not restrict our immune phenotyping study on high-grade serous histology of ovarian cancer, our current study was underpowered to draw firm conclusions on non-serous histology given their relative lower representations in our study. For example, while we found several genetic factors, such as TMB, dMMR and HRD, did not seem to be associated with different immune phenotypes in the ovarian TCGA study (HGS histology only), we cannot rule out that there may be tumour-autonomous genomic/epigenomic drivers of immune infiltration, which will differ between histological types of ovarian cancer. Second, while most of the ovarian tumour tissues analysed in this study are primary tumours, significant portion of patients who are diagnosed with ovarian cancer have a disseminated disease. Several recent reports have shown intra-tumour heterogeneity within the same individual between the metastatic and primary tissues as well as within the same tissue (Jimenez-Sanchez Cell 2017, Zhang et al (2018) Cell 173:1755, Jimenez-Sanchez et al Nature Gen. (2020) 52:582). While no significant genomic variations seem depicted between primary tumours and metastases (Lee et al. Cell Reports 2020), the patterns of T cell infiltration and the composition of the tumour microenvironment can vary between and within patients. Hence, further investigations of immune phenotypes in extended non-serous histology and metastatic cohorts of ovarian cancer are warranted.'

2. The nature of the cell lines.

There is robust and on-going debate as to the utility of many ovarian cancer cell lines (especially SKOV3) since the publication of Domcke et al in Nature Communications in 2013. I think that the authors have used a broad panel of ovarian cancer cells and justify their selection adequately. Clearly, there were experiments on EZH2 inhibition in the original manuscript that have been

Desbois, Udyavar, Ryner et al.

removed as well as discussion on the role of SMARCA4 – I agree with reviewer 2 that these are probably only tangentially related to most ovarian cancer subtypes.

Authors' response:

We appreciate the reviewer's feedback on the nature of ovarian cancer cell lines. With many caveats of ovarian cancer cell lines, we have tried to use some of these cell lines as *in vitro* models to provide additional mechanistic insights in addition to the observations in ovarian patient cohorts.

3. The control vs experimental arm patients from ICON7

The authors have made the logical decision to include the experimental arm tumours in the phenotyping analysis but not in the survival associations – this makes complete sense. However, given what is known about role of VEGF signalling in immune responses, it would be intriguing to see whether bevacizumab treatment influences outcomes in the defined immune subsets. Data were presented at ASCO in 2014 (but not published as far as I am aware) to suggest that there were gene expression subsets that did not derive benefit from the addition of bevacizumab in ovarian cancer.

Authors' response:

We agree with the reviewer that it would be interesting to explore if bevacizumab treatment influences outcomes in the defined immune subsets. Unfortunately, the limitation of this study prevented us from further exploring this interesting question. In our current study, the biomarker evaluable population (BEP) in ICON7 only accounts for ~ 24% of intend to treat population (ITT). As shown below (lower panels and table), in the control arm (chemo), the progression-free survival (PFS) in BEP was representative of PFS in ITT. However, in the treatment arm (bevacizumab), the PFS in the BEP and ITT were significantly different. These bias in the treatment arm also led to the overall treatment effect of bevacizumab observed in the ITT could not be reproduced in the BEP (below, upper panels and table). For these reasons, we have been very careful and not to use any data generated from BEP to interpret the treatment effect by bevacizumab, but only evaluate outcome association in the control arm.

4. HRD status/BRCA status in the ICON7 samples

The authors have analysed 88 genes, including BRCA1/2, in the ICON7 samples, and show no correlation immune phenotype (INF vs EC vs DES). This is an important addition to the manuscript.

Authors' response:

We appreciate the reviewer's positive feedback on the added value of additional genetic analysis in ICON7 samples.

5. Multivariate analysis

This is incredibly important, and the authors have included stage in a multivariate analysis. However, two additional critical clinical factors in any survival analysis in ovarian cancer are age and debulking status. There is widespread evidence that age influences immune responses in all diseases, so I think that age would be an important co-variate in these analyses. ICON7 also collected information on debulking status, and the the ICON7 high risk subset of patients included those with residual disease. Moreover, the original publication from George Coukos (Zhang et al 2003) showed that the only group with improved outcomes by debulking status (optimal vs sub-optimal debulking) were those intra-tumoural T cells, implying a link between immune infiltration and debulkability (see Figure 2F in Zhang et al). Thus, it will be important to include associations between immune subsets and debulking status in ICON7, even as a descriptive analysis.

Author's response:

We thank the reviewer for this great suggestion. In this new revision of our manuscript, we have updated this multivariate analysis by including the age and debulking status as covariates. While

age does not influence the outcome in our analysis, we indeed observed that sub-optimal debulking is significantly associated with poor prognosis.

To note, two patients did not have a debulking status. Hence the number of patients presented in the Figure S3 dropped from 172 to 170 patients.

In the revised manuscript, we have updated the following paragraph in the results section to include these additional parameters:

Page 8: ‘Finally, we performed multivariate analysis to include in several known prognosis factors in ovarian cancer such as stage, age and debulking status. We confirmed that patients with late stage disease (stage III and IV) and sub-optimal debulking status were significantly associated with poor prognosis in the ICON7 cohort. However, the association between excluded immune phenotype and poor prognosis remained significant even after correction of potential effect of these known prognosis factors (Fig. S3).’

Desbois, Udyavar, Ryner et al.

Fig. S3. Association of the tumour-immune phenotypes with PFS in a multivariate analysis correcting for the tumour stage, age and debulking status in the ICON7 cohort (control chemo arm only). Source data are provided as a Source Data file.

2. Reviewer three made several comments

1. Gold standard for defining INF vs EXC vs DES

The reviewer raised an extremely important point – how were tumours allocated into the three IHC groups. I acknowledge the authors' reply that figure 1b shows a continuum of immune cell infiltration states (which is physiologically logical). I note that the classifiers of INF vs EXC vs DES were performed using Definiens software, but the online methods section is not clear on how was this developed nor what manual validation/scoring was performed by a certified pathologist (or even if any manual validation was performed). It would be helpful if the classification parameters that the authors developed were made available so that other investigators could attempt to recreate this work using the same parameters on other sample sets.

I note Figure S7, which is an analysis of INF vs EXC+DES, which is an attempt to recreate other analysis, including the OTTA analysis of 2017, and potentially reinforces the challenges of binary classifiers (TIL vs no TIL).

Overall, I think that the authors have addressed the reviewer comments clearly here.

Author's response:

We greatly appreciate the reviewer's feedback. We indeed attempted to recreate the binary classification TIL vs no TIL to demonstrate our study is not in contradiction with previous studies but bring an additional layer of information.

Additionally, we would like to take the opportunity here to further clarify the classification method. Digital pathology using the Definiens software was used only to calculate and differentiate the CD8+ cells infiltrating the stroma vs the tumour epithelium and not for the classification of the 3 tumour immune phenotypes: infiltrated, excluded and desert. The classification integrated the output of the digital pathology with gene expression data. The digital pathology algorithm ran in Definiens could identify the stroma and epithelium regions based on the size and shape of nuclei. Results were visually confirmed by a pathologist on multiple fields of view (FOV). Unfortunately, Definiens software was recently discontinued as a commercial product. Nevertheless, here we provide further information and pseudo-code that should facilitate recreation of this work.

Original images were scanned in NDPI format at 0.46um/pixel. RGB layers were stored as Layer 1 = Red, Layer 2 = Green, Layer 3 = Blue.

Images are first analysed at low magnification (0.2x).
Layer "filtered" = Gaussian blur (kernel 11x11) on Layer 1
"Tissue" region separated from "Background" at threshold 230 on Layer "filtered".
Objects smaller than 5000 pixel² were classified as the same as surrounding class.
"Tissue" region is tiled using "chessboard segmentation" at scale factor 30.

All tiles were analysed at full magnification (20x).
A check is performed to make sure tissue area occupies at least 100000 pixel² in the full resolution tile. Otherwise the tile is not analysed.
Normalized layers are computed as follows:
Layer "brown" = $3 \times 10^4 \cdot \text{Layer 1} / \text{Layer 3} / (\text{Layer 1} + \text{Layer 2} + \text{Layer 3})$
Layer "blue" = $265 \cdot \text{Layer 3} / (\text{Layer 1} + \text{Layer 2} + \text{Layer 3})$

On image object level "Tissue level"
Layer "filtered blue" = Gaussian blur (kernel 101 x 101) on "blue"

Desbois, Udyavar, Ryner et al.

"Stroma" was separated from "Tumour" based on "filtered blue" with threshold 86.

On a separate image object level "Cell level"

"Nuclei" are separated from "Background" using "mean blue" with threshold 100.

Positive staining nuclei and negative nuclei were separated using "mean brown" with threshold 80.

Stained nuclei with "brightness" (averaged intensity of 3 RGB layers) below 90 were categorized as 'anthracosis' artefact, and not counted as positive cells.

The number and areas of positive nuclei were reported relative to total stroma or epithelial area.

This pseudo-code is now included in the Methods section of the newly revised manuscript.
(Page 25)

2. Is CD8 IHC still not the optimum assay for defining immune status rather than adding the complication of gene expression.

The reviewer raises the important point that every pathology lab in every hospital will be able to undertake CD8 IHC, whilst clinical-grade gene expression analyses are much more challenging to implement. I would also raise the fact that the authors here have developed their own gene expression signature set for their primary analyses, with previous signatures (e.g. Tothill, TCGA) as secondary analyses. A very large OTTA/OCAC gene expression signature for HGSC has just been published online (PMID 32473302) that would also be relevant to discuss as it will be important (not least for management of patients) that an agreed signature be developed, validated and applied prospectively from now on.

Author's response:

We agree with the reviewer that it is important to address how gene expression-based classification tools may be further developed, validated and applied prospectively in clinical setting. In the revised manuscript, we added the following paragraphs in the discussion:

Page 17: 'Although histology-based tumour classification has been widely applied in the clinical setting, given the continuous nature of the distribution of tumour-infiltrating T cells, robustly classifying tumour immune phenotypes based on only histological metrics has been challenging. In this study, we developed a novel digital image analysis algorithm to quantify the quantity and spatial distribution of CD8⁺ T cells in the tumour microenvironment. Coupling this digital pathology algorithm with transcriptome analysis in a large cohort of archival **treatment-naïve** tumour tissues from the ICON7 Phase III clinical trial, we built a random forest machine learning algorithm to classify tumour-immune phenotypes in ovarian cancer. This approach yielded a set of high-dimensional quantitative metrics to define tumour-immune phenotypes. Our study also provided the first proof of concept of classifying tumour-immune phenotypes based on a gene expression classifier. With additional optimization, **validation and prospective testing**, the molecular classifier developed in this study may **enable systematic characterization of tumour-immune phenotypes in large clinical trials of ovarian cancer and potentially other solid tumour types as well.**'

We appreciate the reviewer's comment highlighting this recent work from the OTTA consortium and the opportunity to discuss it here in light of our results. Similar to our study, they developed a gene signature that can be applied on FFPE tumour tissue to identify the patients likely to achieve a 5-year survival. Interestingly, two of their top five genes are also present in our molecular classifier: TAP1 and CXCL9. They are both highlighted by the quantitative metric (R),

Desbois, Udyavar, Ryner et al.

suggesting an increase CD8+ T cell infiltration independently of their distribution. In their conclusion, the OTTA consortium emphasized the importance of the immune system. As similar findings were obtained between different groups including the OTTA consortium, Nelson and Shah work and our study, it will be indeed important to develop a consensus signature integrating the tumour immune phenotypes and evaluate it in prospective settings in order to stratify and propose the best therapeutic strategy to the patients. These gene expression signatures are powerful tools, not only to identify biomarkers but also for a better understanding of the biology of the disease and propose new targets/therapeutics.

3. Histopathology of ICON7 and dMMR

The authors point out that there were some endometrioid cases in ICON7, so the presence of dMMR is explicable.

I would also point out that the authors mis-spell endometrioid in Table S8 – it is endometrioid NOT endometroid.

Author's response:

Thank you for pointing out this misspelling. We now have updated the Figure S8 by replacing it by “endometrioid”. We also corrected the percentage for unknown grade in the chemo arm from ‘0% (n=1)’ to ‘1% (n=1)’.

	CARBOPLATIN PLUS PACLITAXEL CHEMOTHERAPY	CARBOPLATIN PLUS PACLITAXEL CHEMOTHERAPY PLUS BEVACIZUMAB
n	182	188
Histology	% (n)	% (n)
Serous	71 (130)	69 (130)
Clear cell	10 (18)	13 (25)
Endometrioid	9 (16)	5 (9)
Mucinous	2 (3)	1 (2)
Serous/Clear Cell/Endometrioid/ Mucinous/Other Mixed	5 (9)	9 (16)
Endometrioid/Clear cell/Mixed Papillary	1 (1)	2 (3)
Cystadenocarcinoma/Unclassified/ Undifferentiated/Adenocarcinoma	3 (5)	2 (3)
Original cancer	% (n)	% (n)
Ovary (epithelial)	90 (164)	88 (165)
Fallopian tube	2 (4)	3 (5)
Primary peritoneum	5 (10)	7 (14)
Mixed	2 (4)	2 (4)
Age Group	% (n)	% (n)
18-39yr	4 (8)	3 (5)
40-64yr	72 (131)	76 (144)
>=65	24 (43)	21 (39)
FIGO Stage	% (n)	% (n)
Stage I	8 (14)	8 (15)
Stage II	11 (20)	14 (26)
Stage III	72 (131)	68 (127)
Stage IV	9 (17)	11 (20)
Grade	% (n)	% (n)
1	5 (10)	4 (8)
2	16 (30)	16 (31)
3	77 (141)	79 (148)
unknown	1 (1)	1 (1)
Platinum Sensitivity	% (n)	% (n)
Sensitive	49 (89)	62 (117)
Intermediate	19 (35)	20 (37)
Resistant	27 (50)	16 (30)
Refractory	1 (2)	0 (0)
NA	3 (6)	2 (4)
Baseline CA-125 Category	% (n)	% (n)
< 2x ULN	49 (89)	39 (74)
>= 2x ULN	50 (92)	60 (112)
NA	1 (1)	1 (2)

Fig. S8. Demographic table detailing the histology, origin, age, tumour stage and grade, platinum sensitivity and CA-125 biomarker expression in the ICON7 cohort (n=370). Specimens are split based on the treatment received by the patient. Source data are provided as a Source Data file.

4. Cell lines

See reply to reviewer 2.

5. Whether appropriate literature has been cited

The authors address this point satisfactorily.

6. Numbers of samples analysed

Again, the authors address this point satisfactorily.

7. Primary site vs metastasis

The reviewer raises an extremely important point here that I think the authors have slightly misunderstood. The authors now present data on recurrent disease – the rebuttal quotes the online methods: “An independent validation collection (n=84) was procured from Cureline, Inc (Brisbane, CA, US). These include 55 treatment-naïve primary tumours and 29 paired metastatic tumours collected at disease progression post front line chemotherapy”. However, this is recurrent disease rather than metastatic disease: 70% women with ovarian cancer already have metastatic disease at the time of diagnosis (Stage III or IV disease). The question of how primary tumours may differ from disease that has spread to the omentum and other sites in the peritoneal cavity at the time of diagnosis is vital for the management of newly-diagnosed patients. More importantly, 50% or more of women with advanced disease are now treated with primary/neoadjuvant chemotherapy following a biopsy, almost invariably from the omentum or peritoneal disease and so the only disease that will be assessed will be metastatic. Moreover, there are strong published data to show that primary chemotherapy alters immune cell infiltration (see e.g. PMID 27306793), meaning that assessment on chemotherapy-naïve tissues are required. Thus, for true clinical utility, the classifiers need to be evaluated on a large number of metastatic samples rather than primary tumours only and there needs to be a clear statement in the discussion about potential discrepancies between primary tumour and disease in e.g. the omentum: the work of Sohrab Shah and Brad Nelson (Zhang et al (2018) Cell 173:1755 and also the recently published work from Martin Miller (Jimenez-Sanchez et al Nature Gen. (2020) 52:582 are important here.

Author’s response:

We agree with the reviewer that it is important for future studies to evaluate the classifiers on a large number of metastatic samples. While our ICON7 tumour sample collection includes mostly primary tumours originated from ovary (see Figure S8), our independent validation collection is composed of both primary tumours from treatment-naïve patients and recurrent tumour tissues. All recurrent tumour tissues were collected from metastatic sites including both peritoneal cavity and distant organs.

In this newly revised manuscript, we included the following changes to the results and discussion sections to further clarify these points:

(1) We changed the term ‘metastatic’ to ‘recurrent’ in the Figure S5 and its legend, as well as the text in the results section.

(2) **Page 23**, online method, we added following changes to describe metastatic sites of these recurrent tumours:

“These include 55 treatment-naïve primary tumours and 29 paired recurrent tumours collected at disease progression post front line chemotherapy (25 collected from peritoneal cavity, 4 from colon, liver or appendix).”

(3) **Page 21**, discussion, the following paragraph was added to discuss limitations of this study including both histology and primary vs. metastatic disease:

'It is also worth to note some of the limitations of this study. First, although we did not restrict our immune phenotyping study on high-grade serous histology of ovarian cancer, our current study was underpowered to draw firm conclusions on non-serous histology given their relative lower representations in our study. For example, while we found several genetic factors, such as TMB, dMMR and HRD, did not seem to be associated with different immune phenotypes in the ovarian TCGA study (HGS histology only), we cannot rule out that there may be tumour-autonomous genomic/epigenomic drivers of immune infiltration, which will differ between histological types of ovarian cancer. Second, while most of the ovarian tumour tissues analysed in this study are primary tumours, significant portion of patients who are diagnosed with ovarian cancer have a disseminated disease. Several recent reports have shown intra-tumour heterogeneity within the same individual between the metastatic and primary tissues as well as within the same tissue (Jimenez-Sanchez Cell 2017, Zhang et al (2018) Cell 173:1755, Jimenez-Sanchez et al Nature Gen. (2020) 52:582). While no significant genomic variations seem depicted between primary tumours and metastases (Lee et al. Cell Reports 2020), the patterns of T cell infiltration and the composition of the tumour microenvironment can vary between and within patients. Hence, further investigations of immune phenotypes in extended non-serous histology and metastatic cohorts of ovarian cancer are warranted.'

Additional minor comments

1. Line 129: what does 'sensible' mean in this context?

Author's response:

To clarify, we have switched 'sensible' to 'appropriate'.

'The two-dimensional metrics defining CD8+ T cell quantities and distribution for these 122 samples confirmed that the classifier assigned them to an **appropriate** immune phenotype (Fig. 2d, right panel).' line 129

2. Line 219: I think that the 10th word in this line should be 'observed' rather than 'over served'.

Author's response:

The sentence has been corrected to replace 'over served' by 'observed' as correctly pointed out (now line 220). Thank you for the reviewer for identifying this error.

3. Line 229 – 30: the term 'genetic alterations' is a little broad. The authors have look at mutational burden (from a very small panel of genes), dMMR and HR status. That is a narrow spectrum of alterations, especially in light of the fact that a) only 88 genes were sequenced and b) high grade serous carcinoma is driven by copy number alterations and structural variants.

Author's response:

Desbois, Udyavar, Ryner et al.

We agree with the reviewer this quote was too broad and now have updated the paragraph below with changes highlighted in blue:

Page 11: ‘Based on the RNAseq data, we first predicted the tumour-immune phenotypes for 412 **high-grade serous (HGS)** ovarian tumour samples in the TCGA dataset by applying our 157-gene molecular classifier (Fig. S5a and Table S2). Our analysis revealed an overall absence of significant association between tumour-immune phenotypes and TMB, neo-antigen load, dMMR or homologous recombination deficiency (HRD) in **HGS** ovarian cancer, with an exception that a slightly lower neoantigen load was observed in the desert compared to the infiltrated tumours (Fig. 3f). Together, **our** results suggest that **these** genetic alterations **may** not **be** a major driver of the infiltration or exclusion of CD8+ T cells **in HGS ovarian cancer.**’

4. Line 296: hypermethylation is mis-spelled.

Author’s response:

We have now corrected this misspelling.

5. Line 431: “this study provided the first systematic and in-depth characterization of the molecular features and mechanisms underlying the tumour-immune phenotypes in human cancer.” I think that this statement is perhaps hyperbole – see Zhang et al (2018) Cell 173:1755 and also Jimenez-Sanchez et al Nature Gen. (2020) 52:582 as two examples in ovarian cancer alone.

Author’s response:

Here, we wanted to emphasize the lack of in-depth molecular characterization specific to the tumour immune phenotypes (i.e. infiltrated/inflamed, excluded and desert tumours). But the reviewer is right suggesting toning down this statement. While Jimenez-Sanchez and colleagues work is more focus on the dichotomic view of immune infiltration (hot vs. cold tumour microenvironment), Zhang and colleagues indeed characterized three immune TIL subtypes similar to our tumour-immune phenotypes: N-TILs (similar to desert tumours), S-TILs (similar to excluded tumours) and E-TILs (similar to infiltrated tumours).

In the newly revised version, we modified this statement: ‘In summary, this study provided the first systematic and in-depth characterization of the molecular features and mechanisms underlying the tumour-immune phenotypes in human cancer.’ to ‘In summary, this study provided **an** in-depth **and systemic** characterization of the molecular features and mechanisms underlying the tumour-immune phenotypes in human **ovarian** cancer. (Page 21)

REVIEWERS' COMMENTS:

Reviewer #4 (Remarks to the Author):

The authors have provided robust answers to the issues raised in my previous review (which, in turn, reflected the comments made by reviewers 2 and 3 originally). I am happy with the answers and the changes made to the manuscript.

Point-by-point response to Reviewer's comments

Reviewer #4 (Remarks to the Author):

The authors have provided robust answers to the issues raised in my previous review (which, in turn, reflected the comments made by reviewers 2 and 3 originally). I am happy with the answers and the changes made to the manuscript.

Author's response:

We would like to sincerely thank the reviewer for her/his time in carefully reviewing our manuscript and her/his constructive feedback. We are glad our answers and revised manuscript lead, in principle, to the publication of this manuscript in *Nature Communications*.